# GADBench: Revisiting and Benchmarking Supervised Graph Anomaly Detection

**Jianheng Tang**[1,2][*], **Fengrui Hua**[1], **Ziqi Gao**[1,2], **Peilin Zhao**[3], **Jia Li**[1,2][†]
[1]Hong Kong University of Science and Technology (Guangzhou)
[2]Hong Kong University of Science and Technology, [3]Tencent AI Lab
{jtangbf,zgaoat}@connect.ust.hk, huafengrui@outlook.com,
masonzhao@tencent.com, jialee@ust.hk

## Abstract

With a long history of traditional Graph Anomaly Detection (GAD) algorithms and recently popular Graph Neural Networks (GNNs), it is still not clear (1) how they perform under a standard comprehensive setting, (2) whether GNNs can outperform traditional algorithms such as tree ensembles, and (3) how about their efficiency on large-scale graphs. In response, we introduce GADBench—a benchmark tool dedicated to supervised anomalous node detection in static graphs. GADBench facilitates a detailed comparison across 29 distinct models on ten real-world GAD datasets, encompassing thousands to millions (∼6M) nodes. Our main finding is that tree ensembles with simple neighborhood aggregation can outperform the latest GNNs tailored for the GAD task. We shed light on the current progress of GAD, setting a robust groundwork for subsequent investigations in this domain. GADBench is open-sourced at `https://github.com/squareRoot3/GADBench`.

## 1 Introduction

Graph Anomaly Detection (GAD) is the process of identifying uncommon graph objects, such as nodes, edges, or substructures, that significantly deviate from the majority of reference objects within a graph database [4, 55]. With a notable history spanning over two decades, GAD has proven its effectiveness across a variety of applications. These include, but are not limited to, the prevention of financial fraud [36] and money-laundering [83], the prediction of network intrusions [57, 64] and device failures [46], the identification of spam reviews [56] and fake news [8]. Unlike outlier detection in tabular data, GAD considers the inter-dependencies among a group of objects, which can often yield additional insights for identifying fraudulent patterns. Meanwhile, GAD also presents unique challenges in terms of modeling efficiency and necessitates strategies to address issues such as label imbalance [50], feature heterophily [26], and relation camouflage [22].

The definitions of GAD can be multifaceted, depending on the specific objectives and applications. In this paper, we focus on the most prevalent GAD scenario—**the detection of anomalous nodes within a static attributed graph**. Despite the plenty of methods proposed for this task, including traditional structural pattern mining algorithms [57, 4] and advanced deep learning techniques [21, 55], several limitations exist in the current model development and evaluation scheme:

- **Absence of a comprehensive supervised GAD benchmark.** We summarize the exiting GAD benchmarks and toolboxes in Table 1. The latest benchmark available for GAD, BOND [49], exclusively evaluates unsupervised methods. This overlooks the fact that many GAD models rely on labeled data to boost their performance. In comparison, there have been a variety of supervised

---

[*]Work done during an internship at Tencent AI Lab.
[†]Corresponding Author.

Table 1: Comparison of existing GAD benchmarks in terms of datasets, models, and scenarios.

| Benchmark &Toolbox | #Datasets (Organic) | Max. Nodes /Edges | #Models | Model Type | Supervision Scenario |
|---|---|---|---|---|---|
| UGFraud [23] | 1 (1) | 45K/3M | 6 | GNN | Unsupervised |
| DGFraud [22] | 3 (1) | 45K/3M | 9 | GNN | Supervised |
| BOND [49] | 9 (6) | 3M/4M | 14 | GNN, Classic | Unsupervised |
| GADBench | 10 (10) | 5M/73M | 29 | GNN, Classic, Trees | Fully- and Semi-Supervised |

anomaly detection benchmarks for time series [59, 61], images [68], videos [1], and tabular data [31, 77]. Semi-supervised approach is another setting that calls for attention because it can strike a balance between label annotation budgets and model performance.

- **Insufficient comparative studies between tree ensembles and GNNs.** Tree ensemble methods, including Random Forest [12], XGBoost [16], and Isolation Forest [48], have long been favored in the industry. They also showcase impressive results in a recent benchmark for anomaly detection within tabular datasets [31]. These models are also applicable to GAD datasets [83] with appropriate feature engineering. However, a systematic comparison between tree ensembles and GNNs in the context of GAD is still absent.

- **Limited exploration on large-scale graphs.** While many GNN models tailored for GAD have shown promising results on small-scale datasets, their efficacy on large graphs remains unexplored. On the other hand, although several large-scale GAD datasets have been proposed [36, 83, 74], they mainly focus on the comparisons of GNN variants, making it unclear how they perform compared with traditional GAD algorithms.

To redress these gaps and foster academia-industry synergy in GAD evaluation, we propose GAD-Bench, which serves as the first comprehensive benchmark for supervised GAD. Our evaluation encompasses 7 non-graph models, 10 standard GNNs, and 10 state-of-the-art GNNs specifically designed for GAD. Recognizing the proven success of tree ensembles in the anomaly detection of tabular data, we additionally employ two tree ensemble models with simple neighborhood aggregations. All models are evaluated on 10 real-world GAD datasets ranging from thousands to millions of nodes. They are tested in both semi-supervised and fully-supervised settings, with and without hyperparameter tuning.

Through extensive experiments, we discover that (1) surprisingly, tree ensembles with neighbor aggregation have the superior performance among all models; (2) most standard GNNs are not suitable for the GAD task; (3) GNNs specifically designed for GAD require hyperparameter tuning to achieve satisfactory performance. We highlight that our findings can offer the research community a clearer insight into the current progress in GAD. In summary, our contributions can be organized into three main aspects:

- We introduce GADBench, the first comprehensive benchmark for supervised anomalous node detection on static attributed graphs. This includes a comparison of 29 well-known GAD methods across a collection of 10 real-world datasets in both semi-supervised and fully-supervised settings.

- To ensure a rigorous and fair comparison, we identify the limitations inherent in the existing evaluation scheme for GAD, instituting enhancements from dataset selection, metric utilization, model training, and hyperparameter tuning.

- We integrate all models, datasets, and evaluation protocols mentioned in this paper into an open-source repository. Users can reproduce the results and evaluate their own datasets/models with minimal effort.

## 2 Preliminaries and Related Work

**Task definition.** We focus on the detection of anomalous nodes within a static attributed graph. Formally, consider a graph $\mathcal{G} = \{\mathcal{V}, \boldsymbol{A}, \boldsymbol{X}\}$, where $\mathcal{V} = \{v_1, v_2, \cdots, v_N\}$ denotes a set of $N$ nodes. The non-negative adjacency matrix $\boldsymbol{A} \in \mathbb{R}^{N \times N}$ is defined such that $\boldsymbol{A}_{ij} > 0$ if and only if there is an edge between $v_i$ and $v_j$. $\boldsymbol{X} \in \mathbb{R}^{N \times d}$ is a feature matrix, of which the $i$-th row vector $\mathbf{x}_i = \boldsymbol{X}(i, :)$ is the $d$-dimensional feature vector of node $v_i$. Given a subset of labeled nodes $\mathcal{V}^* \subset \mathcal{V}$, the task is to classify the remaining nodes into either normal or anomalous categories. In the context of

a heterogeneous graph, multiple adjacency matrices $\{A_1, A_2, \cdots\}$ might exist, each representing different types of relationships among nodes.

Although the GAD task can be regarded as a binary node classification problem, it introduces several additional challenges. Firstly, anomalous nodes typically constitute a small portion of the total nodes, resulting in a significant data **imbalance** [50, 74]. Secondly, graphs with anomalies often exhibit strong **heterophily**, where connected nodes possess varying features and labels [27, 26]. This necessitates strategies to handle neighborhood feature disparities during message passing. Lastly, anomalous nodes have a tendency to **camouflage** their features and connections, blending seamlessly by mimicking the normal patterns in the graph [22, 51]. This demands attention to intentionally manipulated edges and node features. Before presenting our benchmark, we provide a brief overview of both classic methods and deep learning based GAD models. For a more thorough review of GAD models, please refer to these relevant surveys [55, 4, 38].

**Classical methods.** Classical methods primarily detect graph anomalies by leveraging graph statistical patterns [57, 34], community and clustering structures [14, 43, 86], egonet information [3], spectral analysis [25], random walk-based techniques [81, 29], etc. Other than these heuristic approaches and handcraft feature engineering, another research line leverages learning methods for a more flexible encoding of graph information to identify anomalies. Examples include residual learning [45, 63], relational learning [40, 66], and Bayesian models [33]. Despite their advantages, most methods are primarily designed for plain graphs and struggle to handle attributed graphs. Additionally, they predominantly emphasize unsupervised settings and are generally inefficient in utilizing node labels.

**GNNs for GAD.** With superb ability to encode both structure and attribute information simultaneously, GNNs have recently gained popularity in mining graph data [76, 39, 85, 30]. To tackle the unique challenges of graph anomalies, such as imbalance, heterophily, and camouflage, several adaptations of standard GNNs have been proposed [20, 96, 24, 53, 52, 21, 79, 54, 90]. On one hand, **spatial** GNNs have primarily been redesigned at the level of their inherent mechanisms, such as message passing and aggregation [99]. For instance, GAS [44] employs a structure-enhanced pre-processing strategy to establish implicit connections between anomalies. CARE-GNN [22] and GraphConsis [51] combat the camouflage behavior by designing camouflage-resistant message passing and aggregation processes. H2-FDetector [72] introduces a novel information aggregation strategy that enables homophilic connections to propagate similar information, while heterophilic connections disseminate differing information. With the consideration of label imbalance, solutions like PC-GNN [50] and DAGAD [47] use imbalance-aware data sampling and graph augmentation to highlight the importance of anomalies during training. On the other hand, **spectral** GNNs provide a fresh viewpoint that associates graph anomalies with high frequency spectral distributions [26]. For example, BWGNN [74] applies Beta kernel to manage higher frequency anomalies via flexible and localized band-pass filters. AMNet [13] captures both low-frequency and high-frequency signals, adaptively integrating signals of varying frequencies.

**Ensemble models.** Tree ensembles, such as Random Forest [12, 9, 6, 5] and XGBoost [16, 78, 97, 58], have shown superior performance in anomaly detection tasks related to tabular data [93, 10, 2, 100]. For example, XGBOD [93], an extension of the XGBoost algorithm specifically designed for anomaly detection, incorporates scores from other models such as Isolation Forest [48] as additional feature and achieves superior performance. Despite their potential, few research has explored the application of tree ensembles in leveraging both structural and feature information simultaneously for GAD. Beyond trees, various base models can be integrated into ensembles for anomaly detection, as suggested by [89]. The use of neighborhood aggregation has been demonstrated to enhance anomaly detection performance, which can be utilized during various stages such as pre-processing [87], model-training [17], and post-processing [88] phases. In this work, we revisit tree ensemble models and demonstrate their potential to outperform GNNs after combining a straightforward structural neighborhood aggregation.

## 3 The Setup of GADBench

In this section, we present a comprehensive overview of the setup for GADBench. We provide the general selection criteria and considerations for models (Section 3.1), datasets (Section 3.2), and other details (Section 3.3).

Table 2: Categorization of all models used in our evaluation.

| | |
|---|---|
| Classic Methods | MLP [67] , KNN [18], SVM [15], RF [12], XGBoost [16], XGBOD [93], NA [88] |
| Standard GNNs | GCN [39], SGC [84], GIN [85], GraphSAGE [30], GAT [76], GT [73], PNA [17] BGNN [37], RGCN [70], HGT [35] |
| Specialized GNNs | GAS [44], DCI [82], PC-GNN [50], GAT-sep [99], BernNet [32], AMNet [13], BWGNN [74], GHRN [26], CARE-GNN [22], H2-FDetector [72] |
| Tree Ensembles with Neighbor Aggregate | RF-Graph, XGB-Graph |

## 3.1 Benchmark Models

Table 2 provides an overview of the 29 models assessed in GADBench. We briefly introduce each model in the following, and provide a more detailed description in Appendix A.

**Classical methods.** We select three basic algorithms for supervised classification: Multi-Layer Perceptron (MLP), $k$-Nearest Neighbors (KNN), and Support Vector Machine (SVM). Additionally, we incorporate three representative decision tree ensembles, including a bagging-based model, Random Forest (RF); a boosting-based model, Extreme Gradient Boosting Tree (XGBoost); and an advanced XGBoost model named Extreme Boosting Based Outlier Detection (XGBOD). We also assess a recent anomaly detection technique, Neighborhood Averaging (NA), which enhances the outlier scores of existing anomaly detectors by averaging them with the scores of neighboring objects.

**Standard GNNs.** We evaluate several standard GNNs that have proven to be effective across diverse graph learning tasks, including Graph Convolutional Network (GCN), Chebyshev Spectral Convolution Network (ChebNet), Graph Isomorphism Network (GIN), Graph Sample and Aggregate (GraphSAGE), Graph Attentional Network (GAT), Graph Transformer (GT), Principle Neighbor Aggregation (PNA), Boosted Graph Neural Network (BGNN). We also consider two widely used heterogeneous GNNs, including Relational Graph Convolution Network (RGCN) and Heterogeneous Graph Transformer (HGT).

**Specialized GNNs.** This group contains GNNs specifically designed for anomaly detection. We evaluate five spatial GNNs including the Graph-based Anti-Spam Model (GAS), Deep Cluster Infomax (DCI), Pick and Choose GNN (PC-GNN), and the GAT with ego- and neighbor-embedding separation (GAT-sep). For spectral GNNs, we evaluate the graph spectral filter via Bernstein Approximation (BernNet), Adaptive Multi-frequency GNN (AMNet), Beta Wavelet GNN (BWGNN), and the Graph Heterophily Reduction Network (GHRN). Additionally, we assess two GNNs optimized for heterogeneous graphs: the CAmouflage-REsistant GNN (CARE-GNN) and the Fraud Detector with Homophilic and Heterophilic Interactions (H2-FDetector).

**Tree ensembles with neighbor aggregation.** Decision tree ensembles have shown their effectiveness in anomaly detection with tabular data [31], prompting us to adapt them for GAD. To incorporate graph structure information, we follow the idea from a subclass of GNNs that independently manage message passing and node feature transformation [84, 98, 91]. Consequently, we devise tree ensembles with Neighbor aggregation that adhere to the following computational paradigm:

$$\boldsymbol{h}_{v_i}^{(l)} = \text{Aggregate}\{\boldsymbol{h}_{v_j}^{(l-1)}|v_i \in \text{Neighbor}(v_j)\}$$

$$\text{Score}(v_i) = \text{TreeEnsemble}([\boldsymbol{h}_{v_i}^0||\boldsymbol{h}_{v_i}^1||\cdots||\boldsymbol{h}_{v_i}^L]).$$

In this scheme, $\boldsymbol{h}_{v_i}^{(0)} = \mathbf{x}_i$ denotes the initial node attributes, and $\boldsymbol{h}_{v_i}^{(l)}$ represents the feature of node $v_i$ after $l$-layers of neighbor aggregation. Aggregate$(\cdot)$ can take on any aggregation function such as mean, max, or sum pooling. Same as [84, 98, 91], the aggregation process is parameter-free. TreeEnsemble$(\cdot)$ can be any tree ensembles that takes the aggregated features as input to predict the anomaly score of node $v_i$. In GADBench, we utilize Random Forest and XGBoost to instantiate two new tree ensemble baselines with neighbor aggregation, referred to as RF-Graph and XGB-Graph.

## 3.2 Benchmark Datasets

In GADBench, we have gathered 10 diverse and representative datasets, as detailed in table 3, which are chosen based on the following criteria:

Table 3: Statistics of all datasets in GADBench including the number of nodes and edges, the node feature dimension, the ratio of anomalous labels, the training ratio in the fully-supervised setting, the concept of relations, and the type of node features. Misc. indicates the node features are a combination of heterogeneous attributes, possibly including categorical, numerical, and temporal information, More details are shown in Appendix B.

| | #Nodes | #Edges | #Feat. | Anomaly | Train | Relation Concept | Feature Type |
|---|---|---|---|---|---|---|---|
| **Reddit[42, 49]** | 10,984 | 168,016 | 64 | 3.3% | 40% | Under Same Post | Text Embedding |
| **Weibo[92, 49]** | 8,405 | 407,963 | 400 | 10.3% | 40% | Under Same Hashtag | Text Embedding |
| **Amazon[56, 22]** | 11,944 | 4,398,392 | 25 | 9.5% | 70% | Review Correlation | Misc. Information |
| **YelpChi[66, 22]** | 45,954 | 3,846,979 | 32 | 14.5% | 70% | Reviewer Interaction | Misc. Information |
| **Tolokers[65]** | 11,758 | 519,000 | 10 | 21.8% | 40% | Work Collaboration | Misc. Information |
| **Questions[65]** | 48,921 | 153,540 | 301 | 3.0% | 52% | Question Answering | Text Embedding |
| **T-Finance[74]** | 39,357 | 21,222,543 | 10 | 4.6% | 50% | Transaction Record | Misc. Information |
| **Elliptic[83]** | 203,769 | 234,355 | 166 | 9.8% | 50% | Payment Flow | Misc. Information |
| **DGraph-Fin[36]** | 3,700,550 | 4,300,999 | 17 | 1.3% | 70% | Loan Guarantor | Misc. Information |
| **T-Social[74]** | 5,781,065 | 73,105,508 | 10 | 3.0% | 40% | Social Friendship | Misc. Information |

- **Organic anomalies.** Datasets in GADBench exclusively contain anomalies that naturally emerge in real-world scenarios, a distinction from previous studies that employ synthetic anomalies for GAD evaluations [21, 49]. These earlier works typically inject artificial node attributes and structures into normal graphs like Cora [71], resulting in anomalies that are relatively straightforward to be identified and obviously different from real-world anomalies.

- **Various domains.** Datasets in GADBench span multiple domains, including social media, e-commerce, e-finance, and crowd-sourcing. As presented in Table 3, the graph edge in each dataset embodies unique relation concepts, which shows a diverse distribution of applications.

- **Diverse scale.** GADBench datasets cover a wide scale, from thousands to millions of nodes. We have consciously excluded datasets with fewer than 5,000 nodes, such as Bitcoin-Alpha [41], Disney, and Books [69].

- **Imbalance ratio.** We have ensured that the number and ratio of anomalies within the datasets meet a specific criteria: each dataset contains more than 100 anomalies, to ensure stable experimental results, and no more than a 25% anomaly ratio, preserving the inherent imbalance nature of GAD. This criterion leads to the exclusion of the Enron dataset [69].

Among the datasets in GADBench, Weibo, Reddit, Questions, and T-Social are designed to identify anomalous accounts on social media platforms. Tolokers, Amazon and YelpChi datasets aim to detect fraudulent workers, reviews and reviewers on crowd-sourcing or e-commerce platforms. T-Finance, Elliptic, and DGraph-Fin concentrate on identifying fraudulent users, illicit entities and overdue loans in financial networks, respectively. For a more comprehensive description of each dataset, please refer to Appendix B.

### 3.3 Other Details

**Data split.** We employ both fully-supervised and semi-supervised settings for model evaluation. In a fully-supervised setting, we preserve pre-existing data splits when available. If such divisions are not provided, we follow the approach suggested by [74], randomly partitioning nodes into three subsets: 40% for training, 20% for validation, and the remaining 40% for testing. For each dataset, the specific training ratio is reported in table 3. The semi-supervised setting typically involves a smaller training ratio, e.g., 1% or 5% in previous studies [22, 74]. However, due to the variance in graph sizes present in GADBench, a fixed training ratio might lead to substantial discrepancies in the scale of training sets. To more accurately mimic real-world semi-supervised scenarios, we standardize the training set across all datasets to include a total of 100 labels—20 positive labels (anomalous nodes) and 80 negative labels (normal nodes). To ensure robustness in our findings, we execute **ten** random splits on each dataset and analyze the average performance of the model.

**Metrics.** According to existing anomaly detection benchmarks [31, 49], we select Area Under the Receiver Operating Characteristic Curve (**AUROC**), Area Under the Prevision Recall Curve (**AUPRC**) calculated by average precision, and the Recall score within top-$k$ predictions (**Rec@K**) as performance metrics for the GAD task. We set $k$ as the number of anomalies within the test set. For all metrics, anomalies are considered as the positive class, and higher scores indicate better model

performance. Among these metrics, AUROC primarily focuses on overall performance and is not sensitive to top-$k$ predictions, Rec@K only cares top-$k$ performance, and AUPRC strikes a balance between the two. Suppose the test set includes 10 anomalies within 1000 data points and a model ranks them from positions $11th$ to $20th$, it would attain an AUROC of 0.99, an AUPRC of 0.33, and a Rec@10 of 0. We also document the **running time** and **memory consumption** of each model.

**Hyperparameter Optimization.** To control the effect of hyperparameter selection and ensure fairness [11], we standardize the evaluation process with and without hyperparameter tuning. Initially, we employ default hyperparameters as stated in the original papers. To ensure fairness in hyperparameter tuning, we then utilize **random search** [7] to optimize hyperparameters. During one trial on each dataset, we randomly select a set of hyperparameters from the predefined search space for each model. For more information about metrics, default hyperparameters, search spaces, and other implementation details, please refer to Appendix C.

## 4 Experimental Results

In this section, we study the experimental results of all the benchmarked models. We first provide a comprehensive comparison of all models, taking into account both default and optimally tuned hyperparameters. Following that, we aim to conduct an in-depth comparison between tree ensembles with neighbor aggregation and GNN-based methods.

### 4.1 Overall Comparison

In Figure 1, we present an overview of model performance across 10 datasets for all metrics, excluding four GNNs that are specific to heterogeneous graphs. In Table 4, we take a close look at the model performance regarding the AUPRC score after hyper-parameter tuning on each dataset. For comprehensive experimental results, please refer to Appendix D. Our key findings include:

**Ensemble trees with neighbor aggregation have superior performance.** As highlighted in Figure 1, XGB-Graph and RF-Graph consistently surpass other compared models across all metrics using default hyperparameters. The performance gap becomes particularly significant in the fully-supervised setting, i.e., XGB-Graph surpasses BWGNN—the best GNN model in this setting—by an absolute average improvement of 2.0% on AUROC, 12.9% on AUPRC, and 9.8% on Rec@K. In the semi-supervised context, RF-Graph presents an absolute average improvement of 2.8% on AUROC, 8.0% on AUPRC, and 3.1% on Rec@K, as compared to GHRN, the best GNN model in this setting. It is important to highlight that the improvement in AUPRC and Rec@K is more pronounced than that in AUROC due to the imbalanced issue, suggesting that RF-Graph and XGB-Graph are more proficient in predicting top-$k$ high-confidence anomalies. Further, as shown in the bottom of Figure 1, tree ensembles with neighbor aggregation not only outperform GNNs in terms of efficiency but also exhibit lower memory consumption. Although the performance gap of different models narrow after hyperparameter tuning as in Table 4, RF-Graph and XGB-Graph still prevail among 6 out of 10 datasets. Accordingly, our observations show the superior effectiveness and efficiency of RF-Graph and XGB-Graph across diverse GAD datasets and scenarios.

**Most standard GNNs prove unsuitable for GAD.** As shown in Figure 1, it becomes clear that the majority of standard GNNs encounter difficulties when dealing with GAD tasks. To illustrate, the performance of GCN and GIN is on par with that of MLP—a method that does not take graph structure information into account. This indicates that standard GNNs often struggle to effectively handle structure camouflage or feature heterophily problems induced by anomalies. In Table 4, while hyperparameter tuning does improve the results of all standard GNNs, they remain subpar compared to methods in other categories. An exception is GraphSAGE, which displays an average absolute improvement of 10.4% on average AUPRC when optimal hyperparameters are used, making it competitive with specialized GNNs. BGNN, while impressive on specific datasets such as T-Social, exhibits poor performance on other datasets. This inconsistency may result from the inherent instability of its joint training scheme, especially when compared to the more stable two-step approach in XGB-Graph and RF-Graph.

**Specialized GNNs require hyperparameter tuning to achieve satisfactory performance.** Generally, specialized GNNs outperform standard GNNs in Figure 1, indicating that GNNs tailored for GAD can indeed enhance anomaly detection capabilities. However, the performance of these

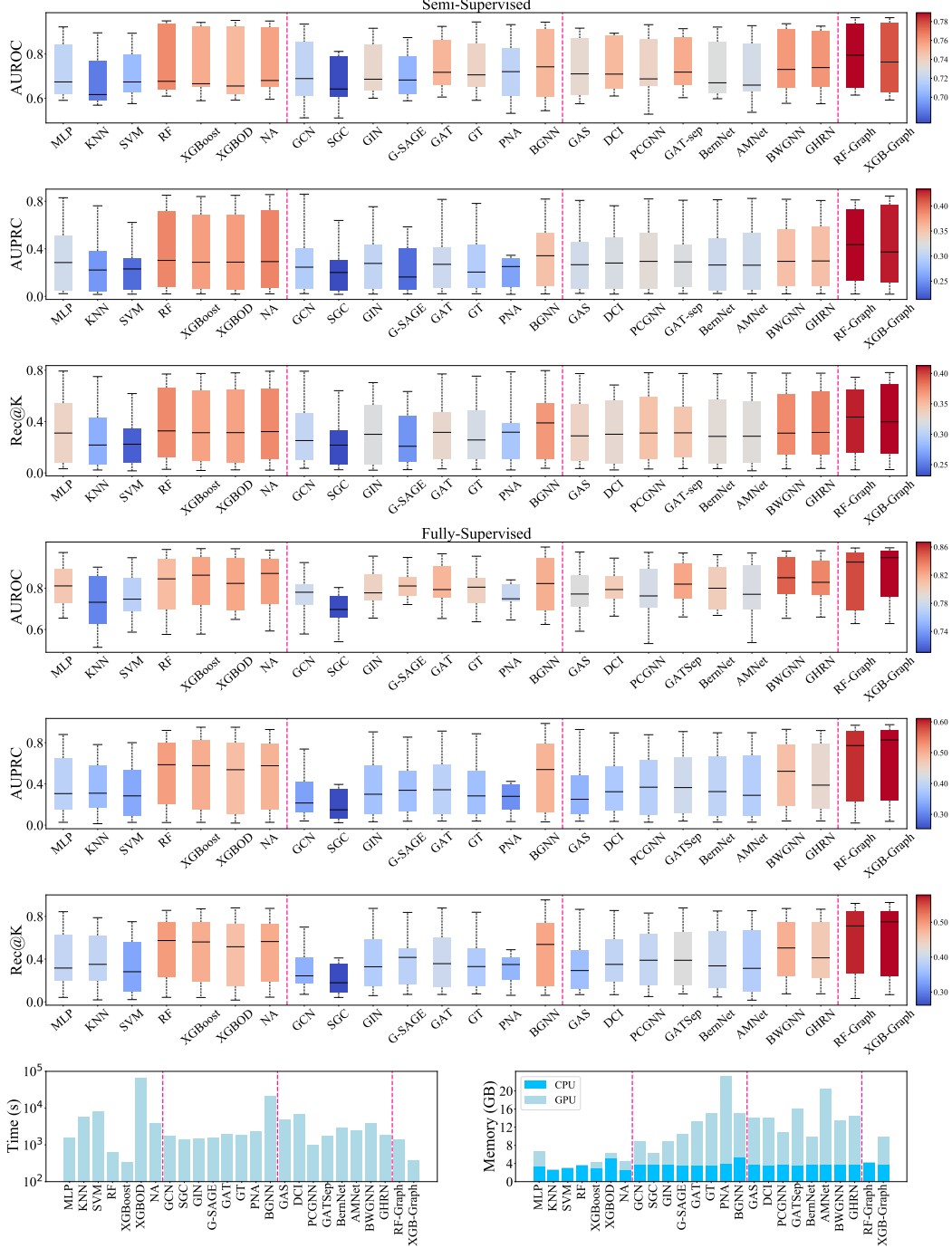

Figure 1: Comparison of the anomaly detection performance, wall-clock time (on all datasets), and peak CPU/GPU memory utilization (on DGraph-Fin) among all models with default hyperparameters. Top three lines are in semi-supervised settings and the others are in fully-supervised settings. The color of the box plot represents the average score for each metric, while the central line within the box indicates the median score.

specialized GNNs strongly depends on hyperparameter tuning. As indicated in the last column of Table 4, all these methods witness a performance improvement after a hyperparameter search. For instance, when optimized hyperparameters are employed, BWGNN can surpass RF-Graph and XGB-Graph on the Reddit dataset. This demonstrates that under certain conditions, some specialized GNNs can deliver commendable performance. However, as illustrated at the bottom of Figure 1, these GNNs often demand more training time and memory. The inherent limitations of hyperparameter

Table 4: Comparison of the AUPRC score of each model with optimal hyperparameters through random search. Best results are highlighted in **bold**. In the last two columns, **Ave.** signifies the average score across the first 9 datasets without T-Social, while **Imp.** denotes the absolute increase in this average score when compared to the default hyperparameters. OOT means the model could not complete training within a day. Results for other metrics can be found in Appendix D and Table 13.

| Model | Reddit | Weibo | Amazon | Yelp. | T-Fin. | Ellip. | Tolo. | Quest. | DGraph. | T-Social | **Ave.** | **Imp.** |
|---|---|---|---|---|---|---|---|---|---|---|---|---|
| MLP | 5.91 | 84.88 | 87.34 | 47.68 | 74.21 | 43.77 | 38.29 | 15.34 | 2.69 | 9.69 | 44.46 | 2.67 |
| KNN | 6.12 | 81.12 | 84.41 | 54.39 | 74.97 | 60.98 | 35.30 | 15.37 | 1.67 | 36.32 | 46.04 | 9.13 |
| SVM | 6.88 | 84.91 | 85.80 | 41.01 | 78.10 | 20.98 | 37.90 | 15.37 | 2.65 | OOT | 41.51 | 4.53 |
| RF | 4.63 | 93.52 | 91.18 | 77.77 | 81.99 | 78.42 | 38.64 | 14.37 | 2.57 | 41.56 | 53.68 | 0.81 |
| XGBoost | 5.56 | 94.49 | 91.88 | 84.00 | 82.64 | 76.93 | 40.05 | 16.24 | 2.75 | 16.60 | 54.95 | 0.73 |
| XGBOD | 8.27 | 95.70 | 92.15 | 79.46 | 82.32 | 74.86 | 40.65 | 16.08 | 1.95 | OOT | 54.61 | 1.62 |
| NA | **9.70** | 94.09 | 91.56 | 63.93 | 88.78 | 29.14 | 51.06 | 14.32 | 4.13 | 79.21 | 49.64 | 3.38 |
| GCN | 4.63 | 94.64 | 45.65 | 20.88 | 78.22 | 25.37 | 40.57 | 14.06 | 3.80 | 76.35 | 36.42 | 1.54 |
| SGC | 6.04 | 91.16 | 42.69 | 19.87 | 68.68 | 17.82 | 39.59 | 10.53 | 2.49 | 16.28 | 33.21 | 5.66 |
| GIN | 6.41 | 91.67 | 84.61 | 33.63 | 78.35 | 26.21 | 40.36 | 13.68 | 3.47 | 60.79 | 42.04 | 2.57 |
| GraphSAGE | 5.56 | 94.02 | 82.45 | 46.64 | 84.71 | 57.82 | 51.41 | 17.50 | 3.77 | 75.32 | 49.32 | 10.44 |
| GAT | 7.20 | 92.91 | 87.94 | 43.62 | 82.72 | 27.53 | 45.25 | 15.51 | 3.85 | 32.07 | 45.17 | 2.80 |
| GT | 7.68 | 89.85 | 84.90 | 44.60 | 83.14 | 25.90 | 45.71 | 17.08 | 3.83 | 36.14 | 44.74 | 5.42 |
| PNA | 7.75 | 96.04 | 35.24 | 29.95 | 76.67 | 27.81 | 41.74 | 11.38 | 3.22 | 21.24 | 36.64 | 4.93 |
| BGNN | 6.87 | 95.99 | 67.92 | 29.71 | 83.72 | 62.03 | 45.35 | 9.43 | **4.24** | **99.09** | 45.03 | 0.95 |
| GAS | 4.43 | 96.76 | 81.43 | 35.11 | 85.95 | 29.80 | 47.21 | 15.48 | 3.65 | 62.36 | 44.42 | 6.62 |
| DCI | 7.74 | 91.77 | 85.17 | 39.88 | 63.68 | 27.39 | 37.73 | 14.59 | 3.31 | 12.97 | 41.25 | 1.01 |
| PCGNN | 7.73 | 89.07 | 89.33 | 44.51 | 83.31 | 42.66 | 44.85 | 15.59 | 3.42 | 80.29 | 46.72 | 4.69 |
| BernNet | 7.82 | 92.38 | 84.89 | 51.92 | 89.17 | 38.25 | 43.69 | 17.25 | 3.27 | 44.30 | 47.63 | 2.90 |
| AMNet | 7.87 | 94.99 | 88.36 | 46.86 | 88.87 | 25.18 | 40.74 | 15.63 | 2.81 | 37.70 | 45.70 | 2.49 |
| GAT-sep | 7.19 | 93.40 | 84.72 | 45.49 | 84.01 | 26.35 | 46.66 | 17.90 | 3.84 | 33.39 | 45.50 | 2.98 |
| BWGNN | 8.32 | 94.01 | 91.48 | 61.53 | 89.23 | 29.31 | 49.58 | **18.57** | 3.97 | 78.93 | 49.57 | 2.12 |
| GHRN | 4.66 | 95.27 | 89.52 | 55.42 | 87.60 | 43.90 | 47.45 | 18.31 | 3.80 | 86.78 | 49.55 | 1.77 |
| RF-Graph | 5.13 | 96.95 | 90.53 | 83.92 | 89.23 | **78.86** | 52.34 | 14.44 | 2.15 | 97.63 | 57.06 | 1.21 |
| XGB-Graph | 5.29 | **97.06** | **93.33** | **91.11** | **90.12** | 77.78 | **53.92** | 18.19 | 3.79 | 97.34 | **58.95** | 1.34 |

search also pose significant challenges, especially in real-world applications where there might be a scarcity of annotated labels or computational resources. Given these constraints, tree ensembles with neighborhood aggregation might still be the preferred choice.

NA is a versatile technique adaptable to any model in GADBench. We apply it to XGBoost and observe a remarkable enhancement in the semi-supervised setting, where the average AUPRC across 10 datasets increases from 37.5% to 38.9%. However, the boost is not significant in the fully-supervised setting. These findings highlight NA's potential as an effective strategy to address challenges associated with label scarcity. For additional results related to the application of NA on other models, please refer to Appendix D.

Finally, we observe that all methods perform poorly on the DGraph-Fin dataset. This can be attributed to the highly imbalanced and sparse graph structure, with an average degree of 1.16. Furthermore, we find that node features in this dataset are highly indistinguishable, as nearly all anomalous nodes share identical features with normal nodes. Indeed, the AUROC scores of all models on this dataset align with those reported in the original paper, as demonstrated in Appendix D.

## 4.2 Specialized Experiments in Heterogeneous and Inductive Settings

Table 5: Comparison of heterogeneous and homogeneous GAD methods on Amazon and Yelp datasets under both fully-supervised and semi-supervised settings. Results are averaged over 10 runs.

| Model | Amazon (Semi-Supervised) | | | Amazon (Fully-Supervised) | | | Yelp (Semi-Supervised) | | | Yelp (Fully-Supervised) | | |
|---|---|---|---|---|---|---|---|---|---|---|---|---|
| | AUROC | AUPRC | Rec@K | AUROC | AUPRC | Rec@K | AUROC | AUPRC | Rec@K | AUROC | AUPRC | Rec@K |
| GAT | 92.44 | 81.57 | 77.07 | 96.66 | 86.67 | 83.10 | 65.56 | 25.03 | 28.08 | 79.50 | 43.41 | 43.65 |
| BWGNN | 91.83 | 81.68 | 77.71 | 97.95 | 89.09 | 85.00 | 64.30 | 23.66 | 26.44 | 84.89 | 55.06 | 52.18 |
| RGCN | 84.17 | 41.07 | 45.57 | 92.03 | 67.97 | 65.49 | 72.20 | 26.46 | 28.54 | 78.34 | 34.57 | 34.98 |
| HGT | 79.75 | 38.13 | 45.07 | 89.64 | 71.46 | 70.22 | 72.83 | 28.49 | 31.59 | 89.62 | 62.63 | 57.75 |
| CARE-GNN | 86.00 | 58.95 | 59.12 | 90.84 | 72.64 | 67.72 | **91.19** | **68.69** | **65.35** | 95.23 | 81.06 | 74.85 |
| H2Detector | 71.00 | 29.27 | 32.61 | 78.66 | 39.95 | 44.35 | 67.28 | 22.23 | 24.08 | 89.07 | 59.40 | 57.54 |
| XGB-Graph | **94.68** | **84.38** | **78.17** | **98.69** | **92.61** | **85.87** | 64.03 | 24.84 | 26.81 | **96.22** | **87.03** | **78.76** |

Table 6: Performance comparison on the Elliptic and DGraph-Fin datasets under inductive and transductive settings, with all results being averaged over 10 runs.

| Model | Elliptic (Inductive) | | | Elliptic (Transductive) | | | DGraph-Fin (Inductive) | | | DGraph-Fin (Transductive) | | |
|---|---|---|---|---|---|---|---|---|---|---|---|---|
| | AUROC | AUPRC | Rec@K | AUROC | AUPRC | Rec@K | AUROC | AUPRC | Rec@K | AUROC | AUPRC | Rec@K |
| GCN | 75.79 | 14.97 | 16.73 | 92.40 | 73.87 | 69.99 | 73.99 | 3.35 | 5.61 | 75.85 | 3.99 | 7.05 |
| GraphSAGE | 79.51 | 19.64 | 20.59 | 82.85 | 34.76 | 45.95 | 72.66 | 3.06 | 5.43 | 75.63 | 3.76 | 6.97 |
| BWGNN | 82.29 | 22.49 | 28.26 | 96.12 | 86.58 | 81.14 | 73.85 | 3.24 | 5.83 | **76.26** | **4.01** | 7.52 |
| GHRN | 84.74 | 25.42 | 28.54 | 96.05 | 86.57 | 81.11 | **76.20** | **4.03** | **7.48** | 76.14 | 3.99 | **7.54** |
| XGB-Graph | **90.36** | **76.20** | **70.64** | **96.80** | **89.58** | **84.59** | 71.23 | 2.81 | 5.33 | 74.64 | 3.66 | 6.75 |

**Dealing with heterogeneous graphs.** We evaluated the performance of four GAD methods that consider heterogeneity—RGCN, HGT, CARE-GNN, and H2-FDetector—using the Amazon and Yelp datasets, each comprising three different types of edges as detailed in Appendix B. For comparison, we also tested three other methods—GAT, BWGNN, and XGB-Graph—that treat all edge types equivalently. As shown in table 5, considering heterogeneity does not enhance performance on the Amazon dataset, but it leads to improvements on the Yelp dataset. Specifically, CARE-GNN outperforms all other methods under label-scarce conditions.

**Performance in the inductive setting.** Our primary experiments focus on the transductive setting, characterized by the assumption that all nodes are visible in the training process. To offer a holistic evaluation, we also conduct experiments in the inductive setting using DGraph-Fin and Elliptic datasets which have temporal features. In this setting, features and structures associated with test nodes are not accessible during the training phase. As presented in table 6, the model performance is generally impacted in the inductive setting of two datasets. Specifically, XGB-Graph outperforms other models across all metrics on Elliptic, while GHRN stands out as the most robust model on DGraph-Fin.

**The impact of different number of neighbor aggregation layers.** Figure 2 illustrates the performance change in XGB-Graph and RF-Graph with varying numbers of neighbor aggregation layers. Observably, the performance on most datasets improves when the number of neighbor aggregation layers increases from 0 to 2, confirming the effectiveness of the neighbor aggregation process. However, further increments in the number of layers do not contribute to any significant improvement in the model performance. Consequently, in most instances, two layers are adequate for XGB-Graph and RF-Graph, and utilizing more layers could lead to unnecessary computational overhead and memory usage.

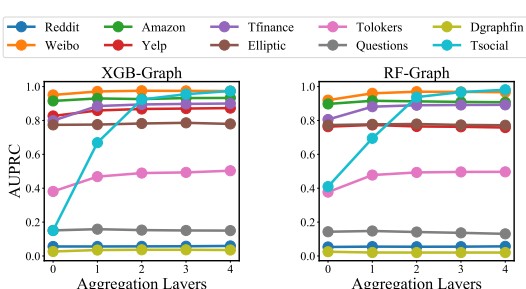

Figure 2: The impact of different number of neighbor aggregation layers on the performance of XGB-Graph and RF-Graph.

### 4.3 Why and When Do Tree Ensembles with Neighbor Aggregation Outperform GNNs?

**An initial study on decision boundaries.** Inspired by a recent benchmark about ensemble trees and neural networks on the tabular data [28], we explore the possible reasons for the superior performance of ensemble trees with neighborhood aggregation. Specifically, our primary investigation focuses on the models' decision boundaries.

In the left panel of Figure 3, we visualize the decision boundaries of GIN and RF-Graph on Amazon dataset. For detailed implementations, please see Appendix E. It is observed that the normal and anomalous nodes are closely intertwined, making them hard to separate. Unfortunately, GIN tends to produce simple and smooth decision boundaries, leading to frequent misclassification of normal nodes in the right bottom corners. Differently, RF-Graph can produce more intricate decision boundaries, demonstrating greater proficiency in distinguishing anomalous data. In the right panel of Figure 3, we visualize the decision boundaries of BWGNN and XGB-Graph on Weibo dataset. As can be seen, the anomalous nodes are grouped into several dispersed clusters. With this dispersed distributions, BWGNN is hard to achieve accurate classification due to simple and continuous decision boundaries. In contrast, XGB-Graph successfully classifies anomalies within each cluster.

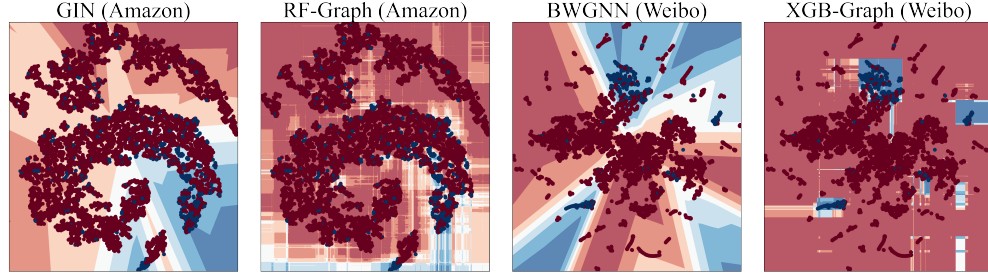

| GIN (Amazon) | RF-Graph (Amazon) | BWGNN (Weibo) | XGB-Graph (Weibo) |

Figure 3: Decision boundaries comparison of different approaches. Blue points represent anomalies while red points are normal nodes. Similarly, the blue/red regions correspond to model predictions for anomalous/normal classes.

In summary, anomaly instances tend to form multiple dispersed clusters and are coupled with normal instances, which fall in the categories of the inductive bias of RF-Graph and XGB-Graph that favor complex and disjoint decision boundaries. In contrast, as GNNs typically employ MLP as the final layer, they tend to generate simple and continuous decision boundaries, which makes GNNs sub-optimal on some challenging GAD datasets.

**The impact of dataset feature types on model performance.** As indicated in Table 3, out of the 10 datasets in GADBench, 3 datasets purely use text embeddings as node features, while in the remaining 7 datasets, node features contain miscellaneous information such as the combination of numerical, categorical, and temporal features. Notably, for datasets that rely on text-based features—namely Reddit, Weibo, and Questions—GNNs showcase competitive performance in comparison to other methods including tree ensembles. This could be attributed to the nature of text embeddings: they often represent low-dimensional manifolds in a high-dimensional feature space, where dimensions tend to be highly correlated. A GNN can process all these dimensions simultaneously, whereas an individual decision tree might only consider a limited subset of feature columns. Conversely, in the other 7 datasets with diverse feature types that have low correlation (for instance, gender and age information), tree ensembles with neighbor aggregation typically exhibit superior performance.

In conclusion, in common GAD scenarios such as fraud detection, node features mainly originate from user profiles, which may encompass varied feature types. Thus, tree ensembles with neighbor aggregation often emerge as the preferred choice. However, for specific tasks like fake news and rumor detection, where text data is pivotal, GNNs still present a compelling option.

## 5   Conclusion and Future Plan

In this paper, we introduce GADBench, the first comprehensive benchmark for supervised anomalous node detection on static attributed graphs. Our evaluation of 29 models on 10 real-world datasets shows that tree ensembles with simple neighborhood aggregation generally outperform other models, including GNNs specifically designed for the GAD task. The rationale behind this finding is initially examined from the standpoints of decision boundary and node feature type. Our results challenge the prevailing belief about the superiority of GNNs in GAD and underline the importance of a fair and comprehensive comparison in accurately understanding the capabilities of various models. By making GADBench open-source, we aim to foster further research and refinement of GAD algorithms, as well as their more informed evaluations and comparisons.

We regard GADBench as a long-term evolving project and are dedicated to its continuous development. Our roadmap for the future includes expanding its scope to include a broader spectrum of GAD scenarios, incorporating more cutting-edge models, and integrating newer datasets. At present, we primarily focus on treating datasets as static graphs to ensure compatibility with most baselines. We have only embarked on preliminary studies concerning heterogeneous and inductive settings. Looking ahead, we envision extending our evaluations to more complex types of graphs and anomalies. Our ultimate goal is to transform GADBench into a more robust, scalable GAD toolbox, with advanced features like automated model selection [95].

**Acknowledgement** This research was supported by NSFC Grant No. 62206067, Tencent AI Lab Rhino-Bird Focused Research Program and Guangzhou-HKUST(GZ) Joint Funding Scheme 2023A03J0673.

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

# A  Detailed Description of Models in GADBench

**Classic Methods**

- **MLP (Multi-Layer Perceptron [67]:** A type of feedforward neural network composed of multiple layers of interconnected artificial neurons that can learn and make predictions by adjusting the weights and biases of the connections between the neurons.
- **KNN ($k$-Nearest Neighbors [18]):** A non-parametric classification algorithm that assigns labels to data points based on the labels of their $k$ nearest neighbors within the feature space.
- **SVM (Support Vector Machine [15]):** A supervised learning algorithm that classifies data by identifying an optimal hyperplane that maximally separates different classes in a high-dimensional feature space.
- **RF (Random Forest [12]):** An ensemble learning algorithm that utilizes bagging to train a collection of individual decision tree models on subsets of the original dataset. The predictions from these individual models are then combined through averaging or voting, resulting in a robust and accurate prediction model.
- **XGBoost (eXtreme Gradient Boosting [16]):** A gradient boosting decision tree framework that uses a gradient descent algorithm to minimize the loss function and iteratively adds new trees into the model to correct the errors made by the previous trees.
- **XGBOD (Extreme Boosting Based Outlier Detection [93]):** An enhanced XGBoost model that integrates unsupervised outlier detectors to generate abnormality scores as additional features for the original data. These generated features are concatenated to the original feature set and used in an XGBoost classifier for outlier detection.
- **Neighborhood Averaging [88]:** A method that post-processes the outlier scores provided by any existing outlier detector by averaging it with the scores of its neighbors in the feature space. In GADBench, we integrated this technique into BWGNN as a case study for in-depth analysis.

**Standard GNN Architectures**

- **GCN (Graph Convolutional Network [39]):** A method that utilizes convolution operation on the graph to propagate information from a node to its neighboring nodes, enabling the network to learn a representation for each node based on its local neighborhood.
- **ChebNet ( Chebyshev Spectral Convolution Network [19]):** A variant of GCN that applies Chebyshev polynomials to approximate the spectral graph convolution operator. This approach allows the model to capture both local and global graph structures, making it scalable for larger graphs.
- **GIN (Graph Isomorphism Network [85]):** A type of GNN that learns to capture the structure of a graph while respecting graph isomorphism. This means it generates identical embeddings for graphs that are structurally identical, regardless of permutations in their node labels.
- **GraphSAGE (Graph Sample and AggregatE [30]:)** A general inductive learning framework that generates node embeddings by sampling and aggregating features from a node's local neighborhood.
- **GAT (Graph Attention Networks [76]):** A GNN framework that incorporates the attention mechanism. It assigns varying levels of importance to different nodes during the neighborhood information aggregation process, allowing the model to focus on the most informative parts.
- **GT (Graph Transformer [73]):** An adaptation of the neural network architecture that applies the principles of the Transformer model to graph-structured data. It uses masks in the self-attention process to leverage the graph structure and enhance model efficiency
- **PNA (Principle Neighbor Aggregation)[17]:** A novel graph neural network architecture combining multiple aggregators with degree-scalers in the neighborhood aggregation process.
- **BGNN (Gradient Boosting Meets Graph Neural Networks) [37]:** An end-to-end framework that trains GBDT and GNN jointly by allowing new trees to fit the gradient updates of GNN.
- **RGCN (Relational Graph Convolution Network) [70]:** A specialized framework derived from Graph Convolutional Networks, tailored for handling relational data.
- **HGT (Heterogeneous Graph Transformer) [35]:** A GNN framework to model Web-scale heterogeneous graphs by designing node- and edge-type dependent parameters for characterizing heterogeneous attention over each edge, thus facilitating the maintenance of dedicated representations for different types of nodes and edges.

**GNNs Specialized for Graph Anomaly Detection**

- **GAS (GCN-based Anti-Spam [44]):** A highly scalable method for detecting spam reviews. It extends GCN to handle heterogeneous and heterophilic graphs and adapts to the graph structure of specific GAD applications using the KNN algorithm.
- **DCI (Deep Cluster Infomax[82]):** A self-supervised learning scheme that decouples node representation learning from classification for anomaly detection. It mitigates inconsistencies between node behavior patterns and label semantics, and captures intrinsic graph properties in concentrated feature spaces by clustering the entire graph into multiple parts.
- **PC-GNN (Pick and Choose Graph Neural Network [50]):** A framework designed for imbalanced GNN learning in fraud detection. It uses a label-balanced sampler to select nodes and edges for training, resulting in a balanced label distribution in the induced sub-graph. Furthermore, it employs a learnable parameterized distance function to select neighbors, filtering out redundant links and adding beneficial ones for fraud prediction.
- **BernNet [32]:** A GNN variant that offers a robust scheme for designing and learning arbitrary graph spectral filters. It uses an order-K Bernstein polynomial approximation to estimate any filter over the normalized Laplacian spectrum of a graph.
- **GAT-sep [99]:** A GNN designed to enhance learning from graph structures under high heterophily. It combines key designs such as ego- and neighbor-embedding separation, higher-order neighborhoods, and intermediate representation combinations.
- **AMNet (Adaptive Multi-frequency Graph Neural Network [13]):** A method designed to capture both low-frequency and high-frequency signals through stacking multiple BernNets, and adaptively combine signals of different frequencies.
- **BWGNN (Beta Wavelet Graph Neural Network [74]):** A method proposed to tackle the 'right-shift' phenomenon of graph anomalies, i.e., the spectral energy distribution concentrates less on low frequencies and more on high frequencies. It employs the Beta kernel to address higher frequency anomalies through multiple flexible, spatial/spectral-localized, and band-pass filters
- **GHRN (Graph Heterophily Reduction Network [26]):** A method that addresses the heterophily issue in the spectral domain of graph anomaly detection. The approach prune inter-class edges to emphasize and delineate the graph's high-frequency components.
- **CARE-GNN (CAmouflage-REsistant GNN) [22]:** A GNN-based fraud detector designed for multi-relation graphs, which is equipped with three modules that enhance its performance against camouflaged fraudsters.
- **H2-FDetector (Fraud Detector with Homophilic and Heterophilic Interactions) [72]** A heterogeneous GNN framework for fraud detection that facilitates the propagation of analogous information through homophilic connections and varied information via heterophilic connection.

# B  Detailed Description of Datasets in GADBench

**Reddit** [42]: This dataset contains a user-subreddit graph, capturing one month's worth of posts shared across various subreddits. Verified labels of banned users are included. The dataset focuses on the 1,000 most active subreddits and the 10,000 most engaged users, leading to a total of 672,447 interactions. Posts were transformed into feature vectors, each representing the Linguistic Inquiry and Word Count (LIWC) categories of the text.

**Weibo** [42]: This dataset features a graph of users and their associated hashtags from the Tencent-Weibo platform, consisting of 8,405 users and 61,964 hashtags. Suspicious activities are defined as two posts made within specific timeframes, such as 60 seconds. Users that engaged in at least five such activities are labeled as "suspicious", while the rest are categorized as "benign". This process yielded 868 suspicious and 7,537 benign users. The raw feature vector is composed of the location of a micro-blog post and bag-of-words features.

**YelpChi** [66]: This dataset is designed to identify anomalous reviews that unfairly promote or demote products or businesses on Yelp.com. The graph includes three types of edges: R-U-R (reviews posted by the same user), R-S-R (reviews for the same product with the same star rating), and R-T-R (reviews for the same product posted in the same month).

**Amazon** [56]: The goal of this dataset is to identify users paid to write fake reviews for products in the Musical Instrument category on Amazon.com. The graph includes three types of relations:

Table 7: Overview of datasets in GADBench with their corresponding node feature types, feature dimension, and detailed descriptions.

| Dataset | Node Feature Type | #Dim. | Detailed Feature Description |
|---|---|---|---|
| Reddit | Text Embedding | 64 | LIWC text embedding for posts |
| Weibo | Text Embedding | 400 | Bag-of-words features from posts |
| Amazon | Misc. Information | 25 | Hand-crafted user features and statistics |
| YelpChi | Misc. Information | 32 | Hand-crafted review features and statistics |
| Tolokers | Misc. Information | 10 | User profile with task performance statistics |
| Questions | Text Embedding | 301 | FastText embeddings for user descriptions |
| T-Finance | Misc. Information | 10 | User profile details such as registration days |
| Elliptic | Misc. Information | 166 | Timestamps and transaction information |
| DGraph-Fin | Misc. Information | 17 | Timestamps and user profiles details |
| T-Social | Misc. Information | 10 | User profile details such as logging activities |

U-P-U (users reviewing at least one same product), U-S-U (users giving at least one same star rating within one week), and U-V-U (users with top-5% mutual review similarities).

**T-Finance** [74]: This dataset aims to find the anomaly accounts in transaction networks. The nodes are unique anonymized accounts with 10-dimension features related to registration days, logging activities and interaction frequency. The edges in the graph represent two accounts that have transaction records. Human experts annotate nodes as anomalies if they fall into categories like fraud, money laundering and online gambling.

**Tolokers** [65]: This dataset is derived from the Toloka crowd-sourcing platform. Nodes represent workers who have participated in at least one of 13 selected projects, and an edge connects two workers if they worked on the same task. The task is to predict which worker has been banned in one of the projects. Node features are based on worker's profile and task performance statistics.

**Questions** [65]: This dataset is collected from the question-answering website Yandex Q. Nodes are users, and an edge connects two users if one user answered the other user's question during a one-year period (September 2021 to August 2022). The dataset focuses on users interested in the topic "medicine". The task is to predict which users remained active on the website at the end of the period. Node features are the mean of FastText embeddings for words in the user description, with an additional binary feature indicating users without descriptions.

**Elliptic** [83]: This dataset includes a graph of over 200,000 Bitcoin transactions (nodes), 234,000 directed payment flows (edges), and 166 node features. The dataset maps Bitcoin transactions to real-world entities associated with licit categories, such as exchanges, wallet providers, miners, and legal services, as well as illicit categories like scams, malware, terrorist organizations, ransomware, and Ponzi schemes.

**DGraph-Fin** [36]: This dataset is a real-world, large-scale dynamic graph provided by the Finvolution Group, representing a social network within the financial industry. In DGraph-Fin, a node represents a Finvolution user, and an edge between two users indicates that one user lists the other as their emergency contact. Anomalous nodes in DGraph-Fin represent users who exhibit overdue behaviors. The dataset comprises over 3 million nodes, 4 million dynamic edges, and more than 1 million extremely unbalanced ground-truth nodes.

**T-Social** [74]: This dataset aims to find the anomaly accounts in social networks. It has the same node annotations and features as T-Finance, while two nodes are connected if they maintain the friend relationship for more than three months. Same as T-Finance, human experts annotate nodes as anomalies if they fall into categories like fraud, money laundering and online gambling.

# C   Other Information in GADBench

## C.1   Metrics

**AUPRC (Area Under the Precision-Recall Curve).** AUPRC is a metric that evaluates the performance of classification models by computing the area beneath the Precision-Recall curve. This curve

illustrates the relationship between precision (i.e., the ratio of true positive predictions to all positive predictions) and recall (i.e., the ratio of true positive predictions to all positive labels) at different threshold levels. AUPRC can be calculated by the weighted mean of precisions at each threshold, where the increase in recall from the previous threshold serves as the weight.

**AUROC (Area Under the Receiver Operating Characteristic Curve).** It evaluates a model's ability to discriminate between positive and negative classes by measuring the area under the ROC curve. The ROC curve plots the true positive rate against the false positive rate for varying decision thresholds. An AUROC of 1 indicates perfect discrimination, while an AUROC of 0.5 suggests that the model is no better than random guessing.

**Rec@K (Recall at $k$).** It is determined by calculating the recall of the true anomalies among the top-$k$ predictions that the model ranks with the highest confidence. We set the value of $k$ as the number of actual outliers in the test dataset. It is noteworthy that in this specific scenario, Rec@K is equivalent to both precision at $k$ and the F1 score at $k$.

**Runtime.** To assess the efficiency of the various models, including both traditional algorithms and neural networks, we measure the runtime as the duration from the beginning to the completion of the experiments. This measurement does not distinguish between computation times on CPUs and GPUs.

**Memory.** We record the peak utilization of both CPU and GPU memory during the entire supervised training phase for each algorithm on the DGraph-Fin dataset. This metric is crucial for assessing the computational resource requirements of each model in practical implementations.

### C.2 Additional Experimental Details

**Implementation Details.** To ensure a comprehensive evaluation and maintain fairness across a broad spectrum of models, we develop an open-source toolkit named GADBench[3]. This toolkit is built on top of Pytorch 1.12 [60] and DGL 1.0 [80]. We implement all standard and specialized GNNs in GADBench using the DGL library. For classic models KNN, SVM, and RF, we use the implementations provided in the Scikit-Learn library [62]. XGBoost is integrated using its official implementation [16], and XGBOD is included via the PyOD library [94].

**Hardware Specifications.** All our experiments were carried out on a Linux server equipped with an AMD EPYC 75F3 32-Core CPU processor, 64GB RAM, and an NVIDIA RTX A6000 GPU with 48G memory.

**Hyperparameter Settings.** Tables 8 and 9 provide a comprehensive list of all hyperparameters used in our random search, complete with their default values and respective distributions or search spaces. For all configurations, we retain the model that yields the best AUPRC score on the validation set and report the corresponding test performance. Due to limited data, we did not perform hyperparameter search for heterogeneous graph neural networks (RGCN, CARE-GNN, and H2-FDetector).

**New Models and Datasets.** GADBench is highly extensible due to its seamless integration with the DGL library. Datasets structured in the `dgl.graph` format, accompanied by binary node labels, can be effortlessly integrated and leveraged by all models within GADBench. In a similar vein, models that accept `dgl.graph` as input and yield an anomaly score for each node can be smoothly assessed across all datasets in GADBench. We are committed to the ongoing maintenance of GADBench and will persist in incorporating new models and datasets.

## D  Additional Experimental Results

In accordance with Figure 1, Tables 11 and 12 provide the performance metrics of each model under semi-supervised and fully-supervised settings, respectively, when default hyperparameters are employed. Additionally, Table 13 consolidates the information in Table 4 and exhibits the performance of the models in terms of AUROC and Rec@K scores subsequent to hyperparameter tuning for each dataset.

In table 10, we evaluated the effect of incorporating Neighborhood Averaging (NA) on the performance of four representative graph-based methods in GADBench, namely GIN, BWGNN, XGB-Graph, and RF-Graph. In the semi-supervised setting, all of the techniques demonstrate an im-

---

[3]`https://github.com/squareRoot3/GADBench`

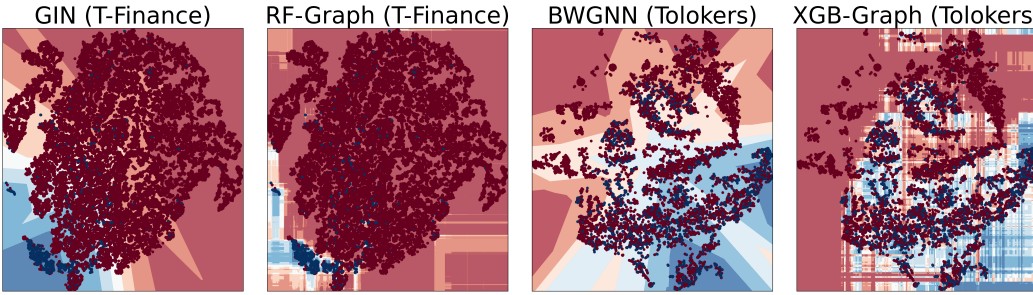

Figure 4: More decision boundaries comparison examples as a complement to Figure 3. Blue points represent anomalies while red points are normal nodes. Similarly, the blue/red regions correspond to model predictions for anomalous/normal classes.

provement in performance subsequent to the integration of NA. For example, the AUPRC score is improved by 1.4% ∼ 3.7%. In the fully-supervised setting, NA can enhance weaker baselines like GIN. However, for stronger baselines, the performance remains similar (on BWGNN) or might even decrease (on XGB-Graph and RF-Graph).

## E  More Details About Plotting Decision Boundaries

As GNNs require graph structure as the input, it is not feasible to directly illustrate their decision boundaries. In the left panel of Figure 3, we use GIN removing the linear layer to embed nodes into the hidden space, following which we employ t-SNE [75] to reduce node embeddings to two dimensions. We then apply MLP and Random Forest to classify these node embeddings, approximating a comparison between GIN and RF-Graph. It is observed that the embeddings of normal and anomalous nodes are entwined, making them hard to separate. Unfortunately, GIN tends to produce simple and smooth decision boundaries, leading to recurring misclassification of normal nodes in the right bottom corners, thus compromising model performance. Conversely, XGBoost can produce more intricate decision boundaries, demonstrating greater proficiency in distinguishing anomalous data.

The right panel of Figure 3 showcases a trained BWGNN model on the Weibo dataset, visualizing the node embeddings preceding the final MLP layer. Employing t-SNE again to reduce the node embeddings, we compare the decision boundaries of MLP and XGBoost on these embeddings, simulating a comparison between BWGNN and XGB-Graph. As shown in the right panel of Figure 3, the embeddings of anomalous nodes are grouped into several dispersed clusters after training. However, the MLP model struggles with accurate classification due to the lack of a simple boundary between normal and anomalous data. In contrast, XGBoost successfully classifies anomalies within each cluster. In Figure 4, we extend our visualization to include two additional datasets, T-Finance and Tolokers. This serves as a complement to Figure 3, and the observations remain consistent.

Table 8: Default hyperparameters and random search space for MLP and GNN models. Elements in '[,]' is randomly selected with equal probability during each trial. RandInt($a$,$b$) returns an random integer between $a$ and $b$ (both included).

| Model | Hyperparameter | Default value | Distribution / Search Space |
|---|---|---|---|
| *Shared hyperparameters for all neural networks* | | | |
| Common | learning rate | 0.01 | $10^{\text{Uniform}(-3,-1)}$ |
| | dropout rate | 0 | [0,0.1,0.2,0.3] |
| | hidden dimension | 32 | [16,32,64] |
| | epochs | 100 | - |
| *Specific hyperparameters for each model* | | | |
| MLP | layers | 2 | [1,2,3,4] |
| | activation | ReLU | [ReLU, LeakyReLU, Tanh] |
| GCN | layers | 2 | [1,2,3] |
| | activation | ReLU | [ReLU, LeakyReLU, Tanh] |
| SGC | number of hops | 2 | [1,2,3,4] |
| | activation | ReLU | [ReLU, LeakyReLU, Tanh] |
| | MLP layers | 1 | [1,2] |
| GIN | layers | 2 | [1,2,3] |
| | aggregation | mean | [sum, mean, max] |
| | activation | ReLU | [ReLU, LeakyReLU, Tanh] |
| GraphSAGE | layers | 2 | [1,2,3] |
| | aggregation | mean | [mean, GCN, pool] |
| | activation | ReLU | [ReLU, LeakyReLU, Tanh] |
| GAT | layers | 2 | [1,2,3] |
| | attention heads | 4 | [1,2,4,8] |
| GT | layers | 2 | [1,2,3] |
| | attention heads | 4 | [1,2,4,8] |
| PNA | layers | 2 | [1,2,3,4] |
| | activation | ReLU | [ReLU, LeakyReLU, Tanh] |
| BGNN | depth | 6 | [4,5,6,7] |
| | iteration per epoch | 10 | [2,5,10,20] |
| | GBDT learning rate | 0.1 | $10^{\text{Uniform}(-2,-0.5)}$ |
| | normalize features | False | [True, False] |
| | activation | ReLU | [ReLU, LeakyReLU, Tanh] |
| GAS | layers | 2 | [1,2,3,4] |
| | number of neighbors | 5 | RandInt(3,50) |
| | distance function | cosine | [euclidean, cosine] |
| DCI | layers | 2 | [1,2,3,4] |
| | pretain epochs | 100 | [20,50,100] |
| | number of clusters | 2 | RandInt(2,30) |
| PC-GNN | layers | 2 | [1,2,3] |
| | under-sample ratio | 0.7 | Uniform(0.01,0.8) |
| | over-sample ratio | 0.3 | Uniform(0.01,0.8) |
| | distance function | cosine | [euclidean, cosine] |
| GAT-sep | layers | 2 | [1,2,3] |
| | attention heads | 4 | [1,2,4,8] |
| BernNet | MLP layers | 2 | [1,2] |
| | orders | 2 | [2,3,4,5] |
| AMNet | layers | 3 | [1,2,3] |
| | orders | 2 | [2,3] |
| | activation | ReLU | [ReLU, LeakyReLU, Tanh] |
| BWGNN | layers | 2 | [1,2,3,4] |
| | MLP layers | 2 | [1,2] |
| | activation | ReLU | [ReLU, LeakyReLU, Tanh] |
| GHRN | layers | 2 | [1,2,3,4] |
| | MLP layers | 2 | [1,2] |
| | deletion ratio | 0.015 | $10^{\text{Uniform}(-2,-1)}$ |

Table 9: Default hyperparameters and random search space for non-deep learning models. Elements in '[,]' is randomly selected with equal probability during each trial. RandInt($a$,$b$) returns an random integer between $a$ and $b$ (both included).

| Model | Hyperparameter | Default value | Distribution / Search Space |
|---|---|---|---|
| SVM | weights | uniform | [uniform, distance] |
| | L2 regularization $C$ | 1 | $10^{\mathrm{Uniform}(-1,1)}$ |
| RF | number of estimators | 100 | RandInt(10,200) |
| | split criterion | gini | [gini, entropy] |
| | max samples | 1 | Uniform(0.1,1) |
| XGBoost | number of estimators | 100 | RandInt(10,200) |
| | learning rate $\eta$ | 0.3 | $0.5 * 10^{\mathrm{Uniform}(-1,0)}$ |
| | L2 regularization $\lambda$ | 1 | [0,1,10] |
| | subsample rate | 1 | [0.5,0.75,1] |
| | booster | gbtree | [gbtree, dart] |
| XGBOD | Same as XGBoost | | |
| NA | Same as XGBoost, with 1 additional hyperparameter: | | |
| | number of neighbors | 5 | RandInt(0,50) |
| RF-Graph | Same as RF, with 2 additional hyperparameters: | | |
| | aggregation layers $L$ | 2 | [1,2,3,4] |
| | aggregation function | mean | [sum, mean, max] |
| XGB-Graph | Same as XGBoost, with 2 additional hyperparameters: | | |
| | aggregation layers $L$ | 2 | [1,2,3,4] |
| | aggregation function | mean | [sum, mean, max] |

Table 10: The impact of introducing neighborhood averaging (NA) on the performance of four representative methods in GADBench.

| Model | Semi-Supervised | | | Fully-Supervised | | |
|---|---|---|---|---|---|---|
| | AUROC | AUPRC | Rec@K | AUROC | AUPRC | Rec@K |
| GIN | 73.61 | 30.26 | 32.14 | 79.97 | 36.54 | 37.52 |
| with NA | **74.78** | **33.91** | **34.18** | **80.43** | **39.03** | **40.01** |
| BWGNN | 76.00 | 35.42 | **37.36** | **84.73** | **48.19** | 48.01 |
| with NA | **76.95** | **36.96** | 37.34 | 84.70 | 47.93 | **48.08** |
| XGB-Graph | 77.74 | 42.91 | 41.29 | **86.71** | **61.09** | **57.82** |
| with NA | **78.41** | **44.30** | **42.07** | 86.05 | 59.58 | 57.41 |
| RF-Graph | 78.94 | 43.37 | 40.83 | **85.35** | **59.65** | **56.90** |
| with NA | **79.56** | **45.14** | **42.28** | 85.17 | 58.54 | 55.33 |

Table 11: Comparison of AUPRC (top), AUROC (middle), and Rec@K (bottom) for each model employing default hyperparameters in the **semi-supervised** setting. Each model is executed 10 times with varying random seeds, and the mean scores along with standard deviations are reported. **Ave.** denotes the average score across all 10 datasets.

| AUPRC | Reddit | Weibo | Amazon | Yelp. | T-Fin. | Ellip. | Tolo. | Quest. | DGraph. | T-Social | Mean |
|---|---|---|---|---|---|---|---|---|---|---|---|
| MLP | 4.4±0.8 | 56.2±6.0 | 83.0±1.6 | 23.6±2.3 | 53.4±13.9 | 45.0±5.9 | 33.3±2.1 | 7.7±1.5 | 2.3±0.2 | 3.9±0.8 | 31.3 |
| KNN | 4.0±0.4 | 41.1±7.2 | 76.2±1.2 | 19.3±1.7 | 49.1±6.5 | 25.1±3.7 | 28.3±1.4 | 4.7±1.1 | 1.7±0.1 | 3.8±0.4 | 25.3 |
| SVM | 4.7±0.5 | 44.1±10.0 | 62.2±21.5 | 25.6±2.0 | 20.6±10.8 | 26.5±4.2 | 33.6±1.6 | 7.1±1.6 | 1.9±0.3 | 3.6±0.6 | 23.0 |
| RF | 4.3±0.4 | 61.6±10.0 | 81.9±4.2 | 27.4±3.5 | 74.9±1.5 | 85.0±1.9 | 33.1±2.3 | 11.3±3.6 | 2.1±0.2 | 7.0±1.3 | 38.9 |
| XGBoost | 4.1±0.4 | 62.1±7.6 | 84.0±1.8 | 26.4±4.2 | 70.9±5.0 | 79.5±2.3 | 31.0±1.9 | 9.1±3.4 | 2.0±0.2 | 6.1±1.3 | 37.5 |
| XGBOD | 4.3±0.6 | 64.8±6.6 | 85.0±3.2 | 26.8±3.8 | 70.3±4.4 | 78.4±1.7 | 30.9±2.4 | 9.0±3.2 | 2.0±0.2 | 4.8±1.0 | 37.6 |
| NA | 4.1±0.4 | 69.3±4.4 | 85.6±1.6 | 27.3±3.8 | 73.9±1.5 | 79.2±3.3 | 31.2±2.0 | 10.1±3.5 | 2.1±0.1 | 6.4±1.3 | 38.9 |
| GCN | 4.2±0.8 | 86.0±6.7 | 32.8±1.2 | 16.4±2.6 | 60.5±10.8 | 43.1±4.6 | 33.0±3.6 | 6.1±0.9 | 2.3±0.2 | 8.4±3.8 | 29.3 |
| SGC | 3.8±0.7 | 63.9±7.4 | 29.5±2.6 | 16.1±1.9 | 31.3±18.6 | 24.2±2.1 | 32.2±3.7 | 6.1±0.7 | 1.7±0.2 | 4.7±1.1 | 21.4 |
| GIN | 4.3±0.6 | 67.6±7.4 | 75.4±4.3 | 23.7±5.4 | 44.8±7.1 | 40.1±3.2 | 31.8±3.2 | 6.7±1.1 | 2.0±0.1 | 6.2±1.7 | 30.3 |
| GraphSAGE | 4.5±0.6 | 58.5±6.2 | 42.5±6.1 | 20.9±3.5 | 11.7±5.2 | 43.1±5.6 | 34.0±2.1 | 5.5±1.3 | 2.0±0.2 | 7.8±1.3 | 23.0 |
| GAT | 4.7±0.7 | 73.3±7.3 | 81.6±1.7 | 25.0±2.9 | 28.9±8.6 | 44.2±6.6 | 33.0±2.0 | 7.3±1.2 | 2.2±0.2 | 9.2±2.0 | 30.9 |
| GT | 4.3±0.7 | 78.2±4.7 | 71.6±5.5 | 23.7±3.0 | 17.2±10.9 | 46.4±8.0 | 34.5±2.0 | 7.2±1.6 | 2.2±0.3 | 8.6±1.5 | 29.4 |
| PNA | 4.0±0.5 | 83.4±4.0 | 34.6±3.8 | 16.8±2.7 | 27.4±9.7 | 31.3±2.9 | 32.6±2.3 | 5.7±1.2 | 1.8±0.2 | 22.8±5.5 | 26.0 |
| BGNN | 4.7±0.5 | 81.8±6.3 | 39.0±5.9 | 17.6±1.3 | 75.8±2.2 | 35.3±5.3 | 33.1±3.6 | 5.7±0.7 | 2.0±0.2 | 58.2±14.6 | 35.3 |
| GAS | 4.7±0.7 | 65.7±8.4 | 80.7±1.7 | 21.7±3.3 | 45.7±13.4 | 46.0±4.9 | 31.7±3.0 | 6.3±2.0 | 2.5±0.2 | 8.6±2.4 | 31.4 |
| DCI | 4.3±0.4 | 76.2±4.3 | 72.5±7.9 | 24.0±4.8 | 51.0±7.2 | 43.4±4.2 | 32.1±4.2 | 6.1±1.3 | 2.0±0.2 | 7.4±2.5 | 31.9 |
| PCGNN | 3.4±0.5 | 69.3±9.7 | 81.9±1.9 | 25.0±3.5 | 58.1±11.3 | 40.3±6.6 | 33.9±1.7 | 6.4±1.8 | 2.4±0.4 | 8.0±1.6 | 32.9 |
| GAT-sep | 4.6±0.7 | 76.5±5.8 | 80.9±2.6 | 24.4±3.7 | 34.2±10.1 | 46.8±8.1 | 33.6±2.2 | 7.4±1.3 | 2.4±0.3 | 10.4±1.9 | 32.1 |
| BernNet | 4.9±0.3 | 66.6±5.5 | 81.2±2.4 | 23.9±2.7 | 51.8±12.4 | 40.0±4.1 | 28.9±3.5 | 6.7±2.1 | 2.5±0.2 | 4.2±1.2 | 31.1 |
| AMNet | 4.9±0.4 | 67.1±5.1 | 82.4±2.2 | 23.9±3.5 | 60.2±8.2 | 33.3±4.8 | 28.6±1.5 | 7.4±1.4 | 2.2±0.3 | 3.1±0.3 | 31.3 |
| BWGNN | 4.2±0.7 | 80.6±4.7 | 81.7±2.2 | 23.7±2.9 | 60.9±13.8 | 43.4±5.5 | 35.3±2.2 | 6.5±1.7 | 2.1±0.3 | 15.9±6.2 | 35.4 |
| GHRN | 4.2±0.6 | 77.0±6.2 | 80.7±1.7 | 23.8±2.8 | 63.4±10.4 | 44.2±5.7 | 35.9±2.0 | 6.5±1.7 | 2.3±0.3 | 16.2±4.6 | 35.4 |
| RF-Graph | 4.5±0.4 | 73.8±9.5 | 70.7±5.1 | 23.6±2.5 | 81.1±2.7 | 80.8±3.1 | 35.8±2.4 | 10.1±2.8 | 2.0±0.2 | 51.3±6.2 | **43.4** |
| XGB-Graph | 4.1±0.5 | 75.9±6.2 | 84.4±1.1 | 24.8±3.1 | 78.3±3.1 | 77.2±3.2 | 34.1±2.8 | 7.7±2.1 | 1.9±0.2 | 40.6±7.6 | 42.9 |

| AUROC | Reddit | Weibo | Amazon | Yelp. | T-Fin. | Ellip. | Tolo. | Quest. | DGraph. | T-Social | Ave. |
|---|---|---|---|---|---|---|---|---|---|---|---|
| MLP | 59.1±4.3 | 66.6±7.8 | 92.2±2.3 | 64.7±3.2 | 89.9±1.4 | 89.4±1.4 | 68.1±2.5 | 61.2±3.1 | 69.1±1.2 | 59.1±5.2 | 71.9 |
| KNN | 58.9±2.3 | 72.5±3.3 | 89.5±0.9 | 60.3±3.3 | 87.7±0.9 | 78.6±4.7 | 63.0±1.9 | 58.6±2.3 | 59.8±2.4 | 56.9±2.8 | 68.6 |
| SVM | 61.8±2.4 | 69.6±4.2 | 89.4±4.7 | 65.8±3.0 | 83.4±9.9 | 83.0±3.7 | 68.9±1.1 | 62.0±2.9 | 65.2±3.1 | 57.6±5.4 | 70.7 |
| RF | 61.0±2.6 | 94.1±1.3 | 94.9±0.8 | 68.0±3.8 | 93.0±0.9 | 94.4±1.9 | 66.5±2.1 | 61.5±4.0 | 63.3±4.5 | 67.3±3.6 | 76.4 |
| XGBoost | 58.9±2.5 | 93.8±0.8 | 94.2±1.0 | 67.0±5.1 | 92.4±1.1 | 91.6±1.9 | 65.1±1.3 | 59.5±5.5 | 66.1±2.2 | 65.7±2.8 | 75.4 |
| XGBOD | 59.1±3.1 | 94.3±0.5 | 95.2±0.8 | 66.8±5.2 | 92.5±1.0 | 91.6±1.7 | 64.1±2.3 | 60.6±3.9 | 64.2±2.1 | 61.0±4.6 | 74.9 |
| NA | 59.5±2.3 | 93.7±0.6 | 94.9±1.3 | 69.3±3.6 | 92.0±1.2 | 90.9±2.3 | 65.4±1.2 | 59.7±5.6 | 66.8±1.4 | 64.9±3.1 | 75.7 |
| GCN | 56.9±5.9 | 93.5±6.6 | 82.0±0.3 | 51.2±3.7 | 88.3±2.5 | 86.2±1.9 | 64.2±4.8 | 60.0±2.2 | 66.2±2.5 | 71.6±10.4 | 72.0 |
| SGC | 53.6±5.2 | 81.1±6.5 | 80.2±1.5 | 51.1±2.9 | 74.5±12.3 | 80.6±1.2 | 63.6±5.1 | 63.6±5.0 | 59.5±3.3 | 64.8±5.6 | 67.3 |
| GIN | 60.0±4.1 | 83.8±8.3 | 91.6±1.7 | 62.9±7.3 | 84.5±4.5 | 88.0±2.9 | 66.8±5.2 | 62.2±2.2 | 65.7±1.8 | 70.4±7.4 | 73.6 |
| GraphSAGE | 60.3±3.8 | 81.8±6.5 | 81.4±2.0 | 58.9±4.3 | 68.9±5.5 | 87.4±1.0 | 67.6±4.2 | 61.2±2.9 | 64.8±3.3 | 72.0±2.9 | 70.4 |
| GAT | 60.5±3.9 | 86.4±7.7 | 92.4±1.9 | 65.6±4.0 | 85.0±4.5 | 88.5±2.1 | 68.1±3.0 | 62.3±1.4 | 67.2±1.9 | 75.4±4.8 | 75.1 |
| GT | 59.1±4.4 | 94.4±4.6 | 88.6±2.2 | 64.5±4.3 | 73.5±7.2 | 88.9±1.6 | 69.7±2.5 | 60.7±3.6 | 67.7±2.5 | 71.4±2.1 | 73.9 |
| PNA | 58.4±4.4 | 93.2±1.8 | 79.1±2.9 | 53.2±5.5 | 78.4±9.7 | 83.7±1.1 | 65.7±3.1 | 62.1±2.9 | 60.7±3.9 | 84.4±3.5 | 71.9 |
| BGNN | 63.1±3.3 | 94.4±5.3 | 83.7±2.5 | 54.4±1.2 | 93.7±0.7 | 83.3±2.3 | 65.1±4.1 | 59.8±3.1 | 59.9±3.5 | 93.6±1.4 | 75.1 |
| GAS | 60.6±3.0 | 81.6±1.9 | 91.6±1.9 | 61.1±5.2 | 88.7±1.1 | 89.0±1.4 | 62.7±2.8 | 57.5±4.4 | 69.9±2.0 | 72.1±8.8 | 73.5 |
| DCI | 61.0±3.1 | 89.3±5.3 | 89.4±3.0 | 64.1±5.3 | 88.0±3.2 | 88.5±1.3 | 67.6±7.1 | 62.2±2.5 | 65.3±2.3 | 74.2±3.3 | 75.0 |
| PCGNN | 52.8±3.4 | 83.9±8.1 | 93.2±1.2 | 65.1±4.8 | 92.0±1.1 | 87.5±1.4 | 67.4±2.1 | 59.0±4.0 | 68.4±4.2 | 69.1±2.4 | 73.8 |
| GAT-sep | 60.3±4.4 | 87.8±5.7 | 91.4±2.4 | 65.0±4.5 | 86.3±3.9 | 89.3±2.1 | 69.1±2.5 | 61.9±2.3 | 66.1±4.3 | 74.5±3.7 | 75.5 |
| BernNet | 63.1±1.7 | 80.1±6.9 | 92.1±2.4 | 65.0±3.7 | 91.2±1.0 | 87.0±1.7 | 61.9±5.6 | 61.8±6.4 | 69.0±1.4 | 59.8±6.3 | 73.1 |
| AMNet | 62.9±1.8 | 82.4±4.6 | 92.8±2.1 | 64.8±5.2 | 92.6±0.9 | 85.4±1.7 | 61.7±4.1 | 63.6±2.8 | 67.1±3.2 | 53.7±3.4 | 72.7 |
| BWGNN | 57.7±5.0 | 89.8±4.0 | 91.8±2.3 | 64.3±3.4 | 92.1±2.7 | 88.7±1.3 | 68.5±2.7 | 60.2±8.6 | 65.5±3.1 | 77.5±4.3 | 76.0 |
| GHRN | 57.5±4.5 | 91.6±4.4 | 90.9±1.9 | 64.5±3.1 | 92.6±0.7 | 89.0±1.3 | 69.0±2.2 | 60.5±8.7 | 67.1±3.0 | 78.7±3.0 | 76.1 |
| RF-Graph | 61.4±2.4 | 96.3±1.1 | 92.5±1.3 | 61.6±2.7 | 95.0±0.7 | 93.9±2.1 | 70.4±2.3 | 64.7±3.6 | 64.9±3.2 | 88.6±1.6 | **78.9** |
| XGB-Graph | 59.2±2.7 | 96.4±0.7 | 94.7±0.9 | 64.0±3.5 | 94.8±0.6 | 91.9±1.3 | 67.5±3.4 | 61.4±2.9 | 62.4±4.1 | 85.2±1.8 | 77.8 |

| Rec@K | Reddit | Weibo | Amazon | Yelp. | T-Fin. | Ellip. | Tolo. | Quest. | DGraph. | T-Social | Ave. |
|---|---|---|---|---|---|---|---|---|---|---|---|
| MLP | 6.5±2.0 | 53.2±5.2 | 79.3±1.0 | 26.5±2.7 | 59.9±10.2 | 54.5±6.2 | 35.5±2.6 | 12.0±2.1 | 3.4±0.9 | 3.2±2.1 | 33.4 |
| KNN | 5.7±1.6 | 46.4±7.8 | 75.1±2.0 | 12.0±3.1 | 57.9±4.4 | 31.8±5.4 | 31.4±2.1 | 8.3±3.4 | 2.2±0.7 | 5.7±2.2 | 27.6 |
| SVM | 6.4±2.2 | 48.0±5.5 | 61.8±21.4 | 28.5±2.6 | 18.1±18.0 | 26.6±7.0 | 36.1±1.6 | 11.2±3.0 | 1.6±0.5 | 1.6±0.4 | 24.0 |
| RF | 5.6±1.9 | 56.4±6.9 | 73.1±7.8 | 30.5±3.8 | 69.3±1.0 | 77.0±2.6 | 34.9±2.5 | 12.9±3.9 | 2.8±0.6 | 11.5±2.1 | 37.4 |
| XGBoost | 5.0±1.9 | 56.8±4.5 | 77.5±2.9 | 29.5±5.0 | 66.9±2.7 | 72.1±2.5 | 33.1±2.8 | 11.5±3.3 | 1.8±0.9 | 9.2±3.3 | 36.3 |
| XGBOD | 4.9±1.7 | 59.3±3.5 | 77.9±5.5 | 30.2±4.4 | 66.9±3.3 | 70.5±2.0 | 32.6±2.5 | 11.3±3.4 | 2.3±0.7 | 6.8±2.6 | 36.3 |
| NA | 4.6±1.9 | 61.8±2.5 | 79.2±1.4 | 31.0±4.8 | 67.1±1.7 | 72.5±3.1 | 33.3±2.5 | 12.1±3.8 | 2.1±0.6 | 10.6±2.3 | 37.4 |
| GCN | 6.2±2.2 | 79.2±4.3 | 36.9±2.6 | 16.9±3.0 | 60.6±7.6 | 49.7±4.2 | 33.4±3.5 | 9.8±1.2 | 3.6±0.4 | 10.2±8.1 | 30.6 |
| SGC | 5.9±2.9 | 64.1±7.6 | 33.6±2.3 | 16.4±2.2 | 35.8±19.4 | 26.9±4.4 | 32.7±3.2 | 10.0±1.4 | 2.4±0.8 | 4.0±1.7 | 23.2 |
| GIN | 4.8±1.9 | 66.5±7.3 | 70.4±5.7 | 26.5±6.1 | 54.4±5.0 | 47.6±3.1 | 33.6±3.0 | 10.3±1.1 | 2.1±0.5 | 5.3±2.9 | 32.2 |
| GraphSAGE | 5.8±0.8 | 63.4±6.0 | 48.0±5.6 | 22.9±3.6 | 18.5±9.4 | 48.2±5.8 | 35.2±2.2 | 8.8±2.5 | 2.5±0.7 | 9.5±2.9 | 26.3 |
| GAT | 6.5±2.3 | 70.2±4.6 | 77.1±1.7 | 28.1±3.4 | 36.2±10.3 | 51.4±5.8 | 35.1±1.8 | 10.9±0.9 | 3.1±0.7 | 11.6±3.0 | 33.0 |
| GT | 5.6±1.7 | 75.4±3.3 | 70.2±4.7 | 26.8±3.4 | 24.4±15.7 | 53.1±7.6 | 36.3±1.6 | 10.7±1.6 | 2.8±1.0 | 11.5±3.0 | 31.7 |
| PNA | 4.2±1.9 | 78.7±4.1 | 40.4±3.1 | 17.5±4.5 | 38.6±14.4 | 38.6±3.6 | 34.0±2.2 | 8.4±2.2 | 2.2±0.7 | 29.4±9.3 | 29.2 |
| BGNN | 5.6±2.1 | 79.7±5.3 | 43.2±6.2 | 17.5±1.4 | 72.9±1.9 | 43.4±6.3 | 34.7±2.7 | 9.0±1.5 | 3.6±0.6 | 58.2±12.4 | 36.8 |
| GAS | 6.6±2.5 | 62.0±6.9 | 77.4±1.7 | 24.6±4.1 | 34.2±9.5 | 51.9±5.2 | 33.0±3.9 | 9.1±2.9 | 3.4±0.4 | 11.5±4.6 | 33.4 |
| DCI | 4.5±1.4 | 68.5±3.5 | 68.3±7.2 | 26.8±5.3 | 58.5±6.3 | 50.0±3.8 | 33.5±5.6 | 9.9±1.9 | 2.3±0.7 | 6.3±6.8 | 32.9 |
| PCGNN | 3.0±2.1 | 65.1±6.6 | 78.0±1.5 | 27.8±3.8 | 63.9±6.3 | 46.5±7.3 | 34.3±1.6 | 10.1±3.9 | 3.7±1.0 | 13.5±3.1 | 34.6 |
| GAT-sep | 5.8±2.2 | 71.1±4.9 | 77.5±2.5 | 27.2±4.2 | 43.6±11.7 | 53.9±6.9 | 35.2±2.0 | 11.2±1.9 | 3.4±0.9 | 14.9±2.9 | 34.4 |
| BernNet | 6.4±1.5 | 60.9±4.6 | 77.2±2.1 | 26.8±3.1 | 60.5±11.1 | 47.0±4.5 | 30.1±3.8 | 10.3±2.7 | 3.8±0.6 | 3.3±2.8 | 32.6 |
| AMNet | 6.8±1.5 | 62.1±4.4 | 77.8±2.3 | 26.6±4.3 | 65.7±6.3 | 37.8±6.7 | 30.5±1.9 | 12.7±2.6 | 2.6±0.8 | 1.6±0.5 | 32.4 |
| BWGNN | 6.0±1.4 | 75.1±3.5 | 77.1±1.6 | 26.4±3.2 | 64.9±11.7 | 49.7±6.1 | 35.5±3.1 | 10.9±3.2 | 3.1±0.8 | 24.3±7.4 | 37.4 |
| GHRN | 6.3±1.5 | 72.4±2.6 | 77.1±1.3 | 26.9±3.1 | 67.7±4.3 | 50.8±4.8 | 36.1±3.1 | 11.1±3.4 | 3.4±0.7 | 24.6±7.0 | 37.7 |
| RF-Graph | 5.8±1.7 | 65.5±7.8 | 63.5±4.5 | 24.3±2.3 | 74.5±3.2 | 72.3±3.3 | 38.0±2.5 | 12.9±3.0 | 2.4±0.4 | 49.0±5.7 | 40.8 |
| XGB-Graph | 4.9±1.9 | 68.9±5.7 | 78.2±1.5 | 26.8±3.0 | 72.4±3.8 | 68.9±3.7 | 36.6±3.0 | 10.6±2.9 | 2.5±0.7 | 43.0±7.6 | **41.3** |

Table 12: Comparison of AUPRC (top), AUROC (middle), and Rec@K (bottom) for each model employing default hyperparameters in the **fully-supervised** setting. Each model is executed 10 times with varying random seeds, and the mean scores along with standard deviations are reported. **Ave.** denotes the average score across all 10 datasets.

| AUPRC | Reddit | Weibo | Amazon | Yelp. | T-Fin. | Ellip. | Tolo. | Quest. | DGraph. | T-Social | Ave. |
|---|---|---|---|---|---|---|---|---|---|---|---|
| MLP | 6.0±1.1 | 84.8±1.2 | 88.0±2.2 | 47.7±1.7 | 70.5±2.6 | 22.7±4.7 | 38.5±1.1 | 15.2±1.0 | 2.7±0.0 | 14.7±7.8 | 39.1 |
| KNN | 4.8±0.4 | 72.9±1.6 | 78.2±0.0 | 31.5±0.0 | 66.5±1.6 | 30.5±0.0 | 31.8±0.7 | 14.7±1.0 | 1.3±0.0 | 24.8±0.1 | 35.7 |
| SVM | 6.5±0.8 | 72.0±4.0 | 80.1±6.6 | 40.6±0.0 | 58.0±17.4 | 19.6±0.2 | 37.2±0.9 | 16.4±1.3 | 2.6±0.0 | 3.7±0.8 | 33.7 |
| RF | 5.3±0.6 | 92.0±0.8 | 89.7±0.4 | 76.3±0.3 | 80.6±1.3 | 77.3±0.4 | 37.7±1.1 | 14.3±1.0 | 2.5±0.0 | 41.0±0.1 | 51.7 |
| XGBoost | 5.6±0.7 | 95.1±0.7 | 91.5±0.0 | 82.7±0.0 | 79.9±1.1 | 77.4±0.0 | 38.1±1.3 | 15.1±0.9 | 2.7±0.0 | 15.0±0.2 | 50.3 |
| XGBOD | 7.5±1.4 | 95.0±0.4 | 89.4±1.6 | 66.3±0.4 | 81.1±0.6 | 75.9±0.0 | 41.3±1.2 | 18.4±1.7 | 2.0±0.0 | 8.2±0.0 | 48.5 |
| NA | 5.7±0.7 | 92.9±0.9 | 91.1±0.0 | 76.2±0.0 | 79.2±1.3 | 78.6±0.0 | 39.1±1.3 | 15.6±0.9 | 2.7±0.0 | 15.5±0.2 | 49.7 |
| GCN | 6.5±1.4 | 94.5±1.0 | 35.1±1.3 | 21.1±0.6 | 73.9±5.2 | 21.9±2.9 | 44.5±2.2 | 12.5±1.4 | 4.0±0.1 | 14.7±7.8 | 32.9 |
| SGC | 4.6±0.8 | 92.3±1.6 | 33.4±2.9 | 16.9±1.5 | 39.5±24.5 | 12.8±0.7 | 36.0±3.7 | 10.0±0.8 | 2.4±0.1 | 5.2±1.6 | 25.3 |
| GIN | 5.7±0.9 | 90.6±1.3 | 79.2±1.5 | 36.5±1.5 | 64.2±4.6 | 23.5±4.9 | 39.6±1.8 | 12.7±1.2 | 3.3±0.1 | 10.1±1.6 | 36.5 |
| GraphSAGE | 5.5±0.8 | 85.4±2.3 | 69.0±12.5 | 54.0±3.0 | 34.8±10.9 | 32.9±6.0 | 47.9±1.7 | 16.6±0.8 | 3.8±0.1 | 12.7±3.3 | 36.3 |
| GAT | 5.9±1.0 | 91.3±1.3 | 86.7±1.4 | 43.4±4.6 | 63.1±11.0 | 25.2±5.6 | 46.1±1.8 | 15.7±1.3 | 3.9±0.1 | 9.2±2.0 | 39.0 |
| GT | 5.7±0.9 | 88.7±1.9 | 80.4±4.0 | 55.0±5.4 | 31.6±14.7 | 25.1±4.5 | 47.6±1.9 | 15.9±1.4 | 3.9±0.1 | 9.5±2.1 | 36.3 |
| PNA | 5.4±0.4 | 93.6±0.8 | 42.5±8.6 | 30.5±0.5 | 28.5±12.3 | 27.5±4.3 | 42.3±1.7 | 11.9±0.9 | 3.5±0.1 | 26.4±7.3 | 31.2 |
| BGNN | 7.7±0.6 | 96.2±0.8 | 66.4±2.0 | 26.9±1.7 | 82.6±0.9 | 64.2±4.6 | 43.8±1.0 | 5.8±0.4 | 3.1±0.1 | 98.7±0.2 | 49.5 |
| GAS | 7.1±1.7 | 97.5±0.5 | 48.9±5.0 | 22.3±0.4 | 76.9±2.4 | 27.9±6.6 | 45.7±1.6 | 14.8±1.9 | 3.8±0.1 | 9.3±2.4 | 35.0 |
| DCI | 6.1±0.9 | 89.6±1.8 | 81.5±2.2 | 39.5±7.5 | 62.6±5.7 | 25.4±4.7 | 39.9±1.5 | 14.1±1.5 | 3.6±0.1 | 13.8±4.0 | 37.6 |
| PCGNN | 4.2±0.5 | 81.9±1.7 | 87.8±1.9 | 43.7±2.6 | 69.8±8.0 | 35.6±10.2 | 38.1±2.1 | 14.4±0.9 | 2.8±0.0 | 8.7±2.4 | 38.7 |
| GATSep | 6.1±1.0 | 90.9±2.1 | 87.1±1.1 | 54.9±2.8 | 70.3±11.9 | 26.4±4.5 | 46.5±2.2 | 16.5±1.0 | 3.9±0.1 | 10.4±1.9 | 41.3 |
| BernNet | 7.1±1.2 | 89.7±1.9 | 86.4±2.7 | 50.1±1.5 | 72.4±6.9 | 21.6±2.9 | 43.5±1.3 | 15.2±1.1 | 2.9±0.0 | 5.4±1.0 | 39.4 |
| AMNet | 7.3±1.0 | 89.7±2.2 | 87.3±2.1 | 48.8±1.3 | 74.3±2.5 | 14.7±2.9 | 43.2±1.2 | 14.6±1.2 | 2.8±0.0 | 3.1±0.3 | 38.6 |
| BWGNN | 6.9±1.6 | 94.4±0.7 | 89.1±1.6 | 55.1±1.6 | 86.6±0.8 | 26.0±3.5 | 49.7±1.9 | 16.7±1.3 | 4.0±0.1 | 54.9±16.5 | 48.2 |
| GHRN | 7.2±1.7 | 91.8±1.2 | 89.5±1.2 | 56.6±1.7 | 86.6±1.5 | 27.7±6.6 | 49.9±2.1 | 16.7±1.2 | 4.0±0.1 | 16.3±4.5 | 44.6 |
| RF-Graph | 5.4±0.4 | 97.0±0.3 | 91.3±0.1 | 76.7±0.5 | 89.0±0.9 | 77.8±0.3 | 49.3±1.6 | 14.2±1.2 | 2.0±0.0 | 93.8±0.1 | 59.6 |
| XGB-Graph | 5.6±0.5 | 97.5±0.6 | 92.6±0.0 | 87.0±0.4 | 89.6±0.8 | 78.2±0.0 | 48.9±1.2 | 15.3±0.6 | 3.7±0.0 | 92.4±0.1 | **61.1** |

| AUROC | Reddit | Weibo | Amazon | Yelp. | T-Fin. | Ellip. | Tolo. | Quest. | DGraph. | T-Social | Ave. |
|---|---|---|---|---|---|---|---|---|---|---|---|
| MLP | 65.5±2.8 | 90.9±1.3 | 97.3±1.7 | 82.0±0.7 | 91.4±1.1 | 84.2±2.3 | 73.3±0.7 | 69.3±2.3 | 72.5±0.1 | 80.2±7.3 | 80.7 |
| KNN | 56.5±1.2 | 88.6±1.0 | 90.2±0.0 | 73.3±0.0 | 86.6±1.0 | 83.1±0.0 | 67.5±0.9 | 61.0±1.2 | 51.4±0.0 | 73.0±0.1 | 73.1 |
| SVM | 67.8±1.7 | 84.8±3.0 | 94.8±1.3 | 77.2±0.0 | 91.6±1.6 | 84.2±1.2 | 72.2±0.6 | 66.7±1.8 | 72.0±0.0 | 58.7±5.6 | 77.0 |
| RF | 62.0±2.8 | 98.7±0.2 | 97.2±0.3 | 93.3±0.1 | 94.2±0.5 | 90.1±0.9 | 71.7±0.8 | 57.6±1.6 | 69.1±0.1 | 79.0±0.1 | 81.3 |
| XGBoost | 61.6±3.5 | 99.1±0.1 | 97.9±0.0 | 95.2±0.0 | 94.4±0.4 | 91.2±0.0 | 73.0±0.8 | 57.8±1.4 | 71.6±0.0 | 81.4±0.1 | 82.3 |
| XGBOD | 68.6±1.6 | 99.1±0.1 | 98.3±0.3 | 89.4±0.4 | 95.0±0.3 | 93.0±0.0 | 75.2±0.8 | 69.3±1.3 | 64.9±0.0 | 69.2±0.0 | 82.2 |
| NA | 62.4±3.2 | 98.4±0.3 | 97.5±0.0 | 94.0±0.0 | 93.9±0.3 | 92.8±0.0 | 73.5±0.9 | 59.3±1.6 | 71.8±0.0 | 81.4±0.1 | 82.5 |
| GCN | 65.2±4.8 | 98.0±0.5 | 82.4±0.4 | 57.8±0.6 | 92.4±2.3 | 81.4±1.8 | 75.6±1.2 | 70.9±1.3 | 75.9±0.2 | 80.2±7.3 | 78.0 |
| SGC | 54.5±4.2 | 97.9±0.4 | 80.3±2.0 | 54.1±2.4 | 76.6±13.8 | 75.4±1.2 | 68.3±4.6 | 71.0±1.1 | 66.1±0.3 | 66.1±8.7 | 71.0 |
| GIN | 65.5±3.2 | 95.5±0.8 | 93.8±1.3 | 77.0±0.9 | 88.2±4.2 | 82.7±2.0 | 75.1±0.9 | 69.4±1.3 | 74.0±0.2 | 78.4±2.2 | 80.0 |
| GraphSAGE | 62.8±5.1 | 94.9±1.3 | 89.6±5.4 | 85.2±1.0 | 82.8±3.9 | 85.3±0.7 | 79.3±0.9 | 72.1±1.7 | 75.6±0.2 | 79.2±4.0 | 80.7 |
| GAT | 65.3±3.1 | 95.3±1.4 | 96.7±1.0 | 79.5±1.9 | 92.8±1.5 | 84.9±1.9 | 79.1±1.0 | 71.1±1.6 | 75.9±0.2 | 75.4±4.8 | 81.6 |
| GT | 63.7±4.4 | 95.5±1.2 | 92.4±2.8 | 84.5±2.2 | 81.4±6.5 | 85.1±1.5 | 79.6±0.8 | 70.9±1.2 | 75.8±0.1 | 72.1±3.3 | 80.1 |
| PNA | 64.6±1.5 | 97.8±0.5 | 83.2±4.5 | 74.3±0.5 | 74.3±12.0 | 84.0±1.0 | 75.4±1.0 | 71.8±0.9 | 73.4±0.2 | 78.2±11.0 | 77.7 |
| BGNN | 72.2±1.8 | 98.8±0.3 | 92.3±0.5 | 64.6±3.5 | 95.6±0.4 | 90.1±0.5 | 74.4±0.9 | 62.5±1.2 | 68.5±1.3 | 99.9±0.0 | 81.9 |
| GAS | 67.5±5.3 | 97.5±0.5 | 86.1±1.5 | 59.1±0.5 | 93.3±1.3 | 85.6±1.6 | 77.4±1.0 | 69.4±1.5 | 76.0±0.2 | 76.9±3.6 | 78.9 |
| DCI | 66.5±3.3 | 94.2±1.7 | 94.6±0.9 | 77.8±7.8 | 86.8±4.5 | 82.8±1.5 | 75.5±0.9 | 69.2±1.3 | 74.7±0.1 | 80.8±6.0 | 80.3 |
| PCGNN | 53.2±2.1 | 90.2±1.5 | 97.3±0.8 | 79.7±1.5 | 93.3±1.0 | 85.8±1.8 | 72.8±2.0 | 69.9±1.4 | 72.0±0.3 | 69.2±4.4 | 78.3 |
| GATSep | 66.1±2.6 | 96.3±1.2 | 97.0±0.5 | 84.3±0.8 | 93.5±2.0 | 86.0±1.4 | 79.6±1.1 | 69.4±1.9 | 76.0±0.2 | 74.5±3.7 | 82.3 |
| BernNet | 66.8±3.7 | 94.9±1.5 | 96.2±1.4 | 83.0±0.6 | 92.8±1.7 | 83.1±1.4 | 76.9±0.6 | 68.8±2.9 | 73.2±0.1 | 66.8±5.8 | 80.2 |
| AMNet | 68.4±2.4 | 95.3±1.7 | 97.0±1.6 | 82.6±0.5 | 93.7±0.7 | 77.3±2.5 | 76.8±0.7 | 68.1±2.9 | 73.1±0.1 | 53.6±3.3 | 78.6 |
| BWGNN | 65.4±4.3 | 97.3±0.9 | 98.0±0.7 | 84.9±0.7 | 96.1±0.5 | 85.2±1.1 | 80.4±0.9 | 71.8±2.2 | 76.3±0.1 | 92.0±5.2 | 84.7 |
| GHRN | 66.0±4.5 | 96.7±1.1 | 98.1±0.3 | 85.3±0.6 | 96.0±0.8 | 85.4±1.9 | 80.4±0.8 | 71.8±1.9 | 76.1±0.1 | 79.0±2.4 | 83.5 |
| RF-Graph | 62.8±2.4 | 99.5±0.1 | 97.5±0.3 | 93.3±0.2 | 96.7±0.5 | 91.9±0.5 | 81.2±0.7 | 65.8±1.2 | 65.6±0.3 | 99.2±0.0 | 85.4 |
| XGB-Graph | 62.8±3.2 | 99.5±0.1 | 98.7±0.0 | 96.2±0.2 | 96.8±0.4 | 93.4±0.0 | 80.1±0.5 | 65.7±0.8 | 74.6±0.0 | 99.3±0.0 | **86.7** |

| Rec@K | Reddit | Weibo | Amazon | Yelp. | T-Fin. | Ellip. | Tolo. | Quest. | DGraph. | T-Social | Ave. |
|---|---|---|---|---|---|---|---|---|---|---|---|
| MLP | 7.5±2.9 | 79.2±2.1 | 84.4±1.8 | 46.9±1.3 | 67.8±2.6 | 23.9±11.6 | 39.4±1.1 | 19.3±1.7 | 4.2±0.2 | 20.1±10.9 | 39.3 |
| KNN | 9.3±1.6 | 70.3±1.6 | 78.8±0.0 | 29.4±0.0 | 68.1±1.8 | 44.5±0.0 | 35.3±1.4 | 17.4±1.6 | 1.8±0.0 | 34.9±0.2 | 39.0 |
| SVM | 8.0±2.6 | 68.5±3.0 | 75.1±7.8 | 41.7±0.0 | 60.6±13.1 | 15.2±0.7 | 37.0±1.3 | 19.4±1.8 | 4.2±0.2 | 2.1±1.1 | 33.2 |
| RF | 5.9±1.3 | 82.3±2.1 | 85.7±0.4 | 69.4±0.5 | 74.8±1.4 | 72.5±0.3 | 39.5±1.0 | 17.9±1.5 | 4.2±0.0 | 45.3±0.1 | 49.8 |
| XGBoost | 7.5±1.4 | 87.0±1.1 | 86.4±0.0 | 74.8±0.0 | 72.9±1.5 | 72.0±0.0 | 40.1±1.5 | 17.9±1.1 | 4.1±0.0 | 20.0±0.2 | 48.3 |
| XGBOD | 10.9±3.9 | 88.0±0.5 | 83.8±2.0 | 61.0±0.4 | 74.6±0.8 | 69.7±0.0 | 42.2±1.5 | 18.5±1.3 | 1.7±0.0 | 14.0±0.0 | 46.4 |
| NA | 6.9±1.4 | 86.6±1.4 | 87.5±0.0 | 73.1±0.0 | 73.2±1.5 | 72.6±0.0 | 40.3±1.2 | 18.8±1.3 | 4.4±0.0 | 20.3±0.2 | 48.4 |
| GCN | 10.0±3.2 | 90.3±0.8 | 36.9±1.4 | 23.7±0.6 | 70.0±5.2 | 25.0±6.0 | 43.1±2.0 | 16.5±2.0 | 7.1±0.2 | 20.1±10.9 | 34.3 |
| SGC | 7.8±2.9 | 88.2±1.4 | 35.2±4.3 | 19.7±2.6 | 41.1±23.3 | 11.0±1.7 | 35.0±3.4 | 16.1±1.2 | 4.2±0.2 | 4.1±2.7 | 26.2 |
| GIN | 6.8±2.4 | 87.5±1.1 | 73.0±3.5 | 38.4±1.2 | 65.5±4.8 | 27.3±8.3 | 39.0±1.9 | 17.5±1.7 | 5.9±0.2 | 14.4±3.8 | 37.5 |
| GraphSAGE | 7.2±1.9 | 83.8±2.5 | 67.9±8.6 | 51.3±2.1 | 46.0±15.9 | 37.3±4.6 | 46.5±1.6 | 21.2±1.6 | 7.0±0.4 | 15.6±4.4 | 38.4 |
| GAT | 6.9±2.6 | 87.8±1.4 | 83.1±1.3 | 43.6±2.7 | 64.6±5.5 | 27.9±11.3 | 46.6±1.8 | 19.8±1.6 | 7.4±0.2 | 11.6±3.0 | 39.9 |
| GT | 7.7±2.8 | 87.9±1.7 | 77.8±2.8 | 51.1±3.2 | 39.8±11.9 | 26.3±11.1 | 47.4±1.8 | 19.7±1.7 | 7.5±0.2 | 13.2±4.1 | 37.4 |
| PNA | 6.1±2.2 | 89.3±1.1 | 48.8±6.8 | 31.4±0.6 | 36.5±15.8 | 33.5±5.8 | 42.4±1.2 | 17.0±1.8 | 6.6±0.3 | 36.5±8.3 | 34.8 |
| BGNN | 10.7±1.6 | 91.0±0.9 | 63.9±2.8 | 27.5±1.4 | 77.3±0.8 | 63.8±1.3 | 43.7±1.2 | 8.6±1.6 | 6.2±0.4 | 95.4±0.3 | 48.8 |
| GAS | 9.9±4.3 | 86.6±1.5 | 49.5±6.4 | 24.1±0.6 | 72.9±1.7 | 34.6±9.6 | 45.0±1.8 | 18.2±2.3 | 6.8±0.2 | 10.4±4.3 | 35.8 |
| DCI | 7.3±2.1 | 85.5±2.5 | 76.7±4.0 | 40.3±7.1 | 64.0±5.3 | 29.9±6.2 | 40.2±2.0 | 18.5±1.6 | 6.7±0.3 | 20.6±7.2 | 39.0 |
| PCGNN | 5.0±2.4 | 75.9±2.4 | 83.2±2.5 | 44.4±2.3 | 69.8±5.3 | 40.4±12.0 | 37.6±2.0 | 18.5±1.9 | 5.0±0.2 | 14.5±4.9 | 39.4 |
| GATSep | 7.6±2.9 | 87.9±2.8 | 84.0±0.8 | 51.1±1.4 | 70.3±9.0 | 31.3±8.8 | 46.6±2.0 | 20.0±1.8 | 7.5±0.3 | 14.9±2.9 | 42.1 |
| BernNet | 10.5±2.4 | 85.0±2.5 | 83.2±2.9 | 48.7±0.9 | 71.9±5.5 | 23.9±7.7 | 43.5±1.8 | 19.3±2.1 | 4.6±0.1 | 6.0±1.6 | 39.7 |
| AMNet | 10.3±2.9 | 85.3±2.6 | 83.4±1.6 | 47.7±1.4 | 73.1±3.4 | 8.8±6.3 | 42.8±1.5 | 19.9±2.1 | 4.2±0.1 | 1.6±0.5 | 37.7 |
| BWGNN | 10.7±2.8 | 87.4±1.7 | 85.0±2.7 | 52.2±1.3 | 81.1±0.7 | 31.7±6.2 | 48.7±2.1 | 21.4±2.0 | 7.5±0.3 | 54.3±12.4 | 48.0 |
| GHRN | 11.6±3.2 | 86.7±1.8 | 85.9±1.6 | 53.0±1.5 | 81.1±1.4 | 33.3±10.3 | 49.1±1.5 | 21.3±2.4 | 7.5±0.2 | 24.6±7.0 | 45.4 |
| RF-Graph | 5.8±1.4 | 92.2±0.7 | 84.4±0.5 | 69.6±0.6 | 84.0±1.1 | 72.5±0.3 | 51.3±1.6 | 18.4±2.4 | 3.2±0.1 | 87.6±0.1 | 56.9 |
| XGB-Graph | 7.6±1.1 | 93.1±1.1 | 85.9±0.0 | 78.8±0.6 | 84.6±1.1 | 71.2±0.0 | 48.8±1.3 | 16.5±1.3 | 6.7±0.0 | 85.2±0.1 | **57.8** |

Table 13: Comparison of AUROC (top) and Rec@K (bottom) of each model with optimal hyperparameters through random search. The Results for AUPRC is in Table 4. **Ave.** signifies the average score across the first 9 datasets without T-Social. OOT (out-of-time) means the model could not complete random search within a day. Best results are highlighted in **bold**.

| AUROC | Reddit | Weibo | Amazon | Yelp. | T-Fin. | Ellip. | Tolo. | Quest. | DGraph. | T-Social | **Ave.** |
|---|---|---|---|---|---|---|---|---|---|---|---|
| MLP | 60.71 | 92.89 | 96.94 | 81.64 | 92.24 | 88.32 | 73.78 | 60.15 | 72.34 | 73.06 | 79.89 |
| KNN | 62.53 | 93.57 | 94.91 | 84.60 | 92.69 | 88.03 | 72.09 | 64.00 | 58.50 | 77.64 | 78.99 |
| SVM | 68.18 | 92.52 | 95.80 | 77.21 | 92.59 | 85.02 | 72.59 | 64.77 | 72.03 | OOT | 80.08 |
| RF | 60.79 | 98.77 | 97.53 | 93.80 | 94.85 | 91.68 | 74.14 | 54.85 | 70.37 | 79.94 | 81.87 |
| XGBoost | 64.59 | 98.82 | 98.34 | 95.38 | 95.42 | 89.91 | 74.84 | 62.09 | 72.43 | 81.75 | 83.53 |
| XGBOD | 68.39 | 99.24 | 98.20 | 93.96 | 95.39 | 92.26 | 74.55 | 67.53 | 64.88 | OOT | 83.82 |
| NA | 63.86 | 99.03 | 98.23 | 94.70 | 94.85 | 91.27 | 74.87 | 66.20 | 71.73 | 81.95 | 83.86 |
| GCN | 62.04 | 98.84 | 85.17 | 58.62 | 94.62 | 81.69 | 73.80 | 68.20 | 75.51 | 96.63 | 77.61 |
| SGC | 66.95 | 98.58 | 87.84 | 57.66 | 87.97 | 79.52 | 72.36 | 68.11 | 68.63 | 83.67 | 76.40 |
| GIN | 61.82 | 98.65 | 95.63 | 73.77 | 92.71 | 83.01 | 74.57 | 68.08 | 74.16 | 94.09 | 80.27 |
| GraphSAGE | 63.28 | 97.75 | 90.90 | 82.90 | 95.62 | 87.59 | 80.92 | **73.04** | 75.60 | 95.73 | 83.07 |
| GAT | 64.15 | 98.54 | 97.13 | 79.14 | 95.75 | 86.28 | 78.75 | 68.30 | 75.53 | 90.33 | 82.62 |
| GT | 67.39 | 97.23 | 93.37 | 80.32 | 94.39 | 86.10 | 78.94 | 69.62 | 75.71 | 90.79 | 82.56 |
| PNA | 63.45 | 98.78 | 82.14 | 74.39 | 92.47 | 84.86 | 75.17 | 66.32 | 73.39 | 84.02 | 79.00 |
| BGNN | 68.94 | 98.59 | 90.64 | 63.28 | 96.20 | 87.69 | 76.65 | 68.02 | 76.22 | **99.94** | 80.69 |
| KNNGCN | 60.31 | 99.17 | 90.69 | 69.13 | 94.69 | 88.53 | 74.23 | 66.95 | 75.72 | 87.80 | 79.94 |
| GAS | 60.57 | 99.11 | 92.69 | 76.43 | 96.45 | 86.66 | 78.19 | 68.15 | 75.98 | 94.96 | 81.58 |
| DCI | 66.87 | 97.94 | 95.39 | 78.38 | 87.85 | 85.73 | 73.86 | 64.77 | 73.94 | 83.96 | 80.53 |
| PCGNN | 65.41 | 95.14 | 98.01 | 80.82 | 94.03 | 86.50 | 76.63 | 67.66 | 72.76 | 96.91 | 81.88 |
| GATSep | 66.17 | 98.22 | 95.11 | 80.49 | 94.40 | 85.94 | 79.52 | 70.08 | 75.78 | 87.69 | 82.85 |
| BernNet | 69.83 | 98.00 | 95.79 | 83.54 | 96.67 | 87.54 | 77.51 | 68.58 | 73.58 | 93.73 | 83.45 |
| AMNet | 69.60 | 98.32 | 96.92 | 81.90 | 96.38 | 83.66 | 75.16 | 66.05 | 72.99 | 92.50 | 82.33 |
| BWGNN | **70.82** | 98.13 | 98.27 | 87.13 | 96.93 | 87.03 | 80.41 | 70.87 | **76.30** | 96.88 | 85.10 |
| GHRN | 61.02 | 99.18 | 98.29 | 84.60 | 96.46 | 89.50 | 80.08 | 72.16 | 76.13 | 97.12 | 84.16 |
| RFGraph | 65.35 | **99.43** | 96.73 | 95.24 | **97.28** | **93.21** | 81.88 | 64.77 | 67.78 | 99.69 | 84.63 |
| XGBGraph | 64.74 | 99.29 | **98.74** | 97.37 | 97.15 | 91.78 | **82.85** | 71.02 | 75.83 | 99.76 | **86.53** |

| Rec@K | Reddit | Weibo | Amazon | Yelp. | T-Fin. | Ellip. | Tolo. | Quest. | DGraph. | T-Social | **Ave.** |
|---|---|---|---|---|---|---|---|---|---|---|---|
| MLP | 9.52 | 78.39 | 83.15 | 46.62 | 68.93 | 57.43 | 39.88 | 16.99 | 4.04 | 16.86 | 44.99 |
| KNN | 10.20 | 73.20 | 79.35 | 51.92 | 70.04 | 56.60 | 37.85 | 19.18 | 1.98 | 44.08 | 44.48 |
| SVM | 8.84 | 77.81 | 82.07 | 41.77 | 72.40 | 19.21 | 39.72 | 18.08 | 4.04 | OOT | 40.44 |
| RF | 5.44 | 84.15 | 86.41 | 70.23 | 75.59 | **72.76** | 39.72 | 17.81 | 4.21 | 45.48 | 50.70 |
| XGBoost | 6.80 | 87.03 | 86.41 | 75.08 | 76.01 | 72.58 | 41.90 | 16.71 | 4.30 | 21.86 | 51.87 |
| XGBOD | 10.20 | 87.32 | 86.96 | 71.62 | 75.87 | 69.07 | 42.21 | 16.71 | 1.68 | OOT | 51.29 |
| NA | 8.84 | 85.59 | 87.50 | 73.38 | 75.31 | 71.74 | 42.21 | 18.36 | 4.04 | 21.70 | 51.89 |
| GCN | 6.12 | 88.47 | 44.02 | 23.85 | 74.90 | 33.52 | 39.41 | 17.81 | 7.05 | 73.23 | 37.24 |
| SGC | 8.84 | 87.32 | 46.20 | 22.46 | 67.41 | 22.99 | 39.72 | 16.99 | 3.87 | 24.93 | 35.09 |
| GIN | 10.88 | 89.63 | 80.98 | 36.15 | 73.37 | 32.13 | 39.41 | 18.36 | 6.32 | 64.47 | 43.03 |
| GraphSAGE | 7.48 | 90.78 | 78.26 | 47.15 | 78.09 | 56.69 | 48.75 | 21.37 | 6.84 | 73.74 | 48.38 |
| GAT | 10.88 | 89.63 | 82.61 | 44.23 | 79.75 | 37.86 | 44.24 | 17.26 | 7.14 | 42.07 | 45.95 |
| GT | 12.93 | 85.88 | 78.80 | 44.62 | 81.55 | 30.75 | 44.39 | 20.27 | 6.92 | 43.58 | 45.12 |
| PNA | 10.88 | 39.67 | **90.78** | 30.00 | 72.54 | 36.47 | 42.68 | 13.15 | 5.16 | 26.95 | 37.93 |
| BGNN | 9.52 | **93.37** | 64.67 | 30.77 | 77.25 | 59.56 | 45.33 | 12.60 | **7.70** | 96.89 | 44.53 |
| KNNGCN | 5.44 | 89.63 | 71.74 | 33.23 | 75.87 | 41.55 | 40.34 | 18.08 | 7.18 | 48.67 | 42.56 |
| GAS | 4.08 | 91.93 | 80.43 | 38.00 | 79.75 | 37.49 | 47.04 | 18.90 | 6.02 | 64.58 | 44.85 |
| DCI | 8.16 | 89.05 | 80.98 | 40.46 | 71.98 | 35.27 | 37.85 | 17.26 | 5.85 | 18.27 | 42.99 |
| PCGNN | **14.97** | 84.15 | 85.33 | 43.77 | 79.06 | 43.77 | 43.15 | 18.08 | 6.66 | 73.53 | 46.55 |
| GATSep | 9.52 | 88.18 | 80.98 | 44.23 | 81.55 | 30.56 | 47.20 | 21.37 | 7.31 | 40.84 | 45.66 |
| BernNet | 10.88 | 89.34 | 82.61 | 49.69 | 83.63 | 49.77 | 44.70 | 19.73 | 5.55 | 48.23 | 48.43 |
| AMNet | 11.56 | 88.76 | 83.15 | 45.38 | 84.05 | 30.56 | 42.52 | 17.53 | 4.21 | 43.21 | 45.31 |
| BWGNN | 11.56 | 87.90 | 85.87 | 56.69 | 84.19 | 42.47 | 50.31 | 21.64 | 7.57 | 75.78 | 49.80 |
| GHRN | 5.44 | 89.34 | 85.33 | 51.85 | 81.97 | 50.51 | 46.57 | **22.19** | 6.96 | 82.33 | 48.91 |
| RFGraph | 3.40 | 91.93 | 83.15 | 75.31 | 84.05 | 72.58 | 52.18 | 15.89 | 3.22 | 93.58 | 53.52 |
| XGBGraph | 6.12 | 91.93 | 85.87 | **83.15** | **85.02** | 71.93 | **53.43** | 20.55 | 6.96 | 93.53 | **56.11** |

