# OpenReview forum: "GADBench: Revisiting and Benchmarking Supervised Graph Anomaly Detection"
_NeurIPS.cc/2023/Track/Datasets_and_Benchmarks — NeurIPS 2023 Datasets and Benchmarks Poster_

### Official Review · Reviewer_VEyk · 2023-07-20
**A very nice benchmark, but some improvements are needed.**

**Rating:** 7
**Confidence:** 5

**Strengths:**

The paper is overall well-organized and easy to follow.

The evaluation is comprehensive, comparing 23 methods on 10 different datasets. It is valuable for both researchers and practitioners in the future.

The paper proposes to combine tree ensembles and neighbor aggregate, which is proven effective in the experiments.

Section 4.2 provides insightful visualization of the decision boundary of different approaches and explains the reason why tree ensembles are effective.

**Additional Feedback:**

I will consider increasing my rating if the authors properly answer the questions and/or address the concerns.

**Clarity:**

The authors have not clearly explained the difference between heterophily and camouflage mentioned in the paper, which are similar to some extent.

**Correctness:**

The results shown in Table 9 seem problematic. The value in Ave. column is not equal to the average of the AUROC for XGBoost. Also, the results are not even better after the hyperparameter search compared to Table 8. The results of other methods may also have similar problems.

The memory plot in Figure 1 shows a counterintuitive that CPU memory consumption is usually lower than GPU memory consumption. A more detailed explanation of the measurement of memory consumption is required.

In Figure 1 row 4, the title of the y-axis might be AUROC.

**Documentation:**

The authors provide a fair amount of documentation and code implementation to foster accessibility and reproducibility.

**Ethics:**

The copyright of DGraph-Fin dataset is owned by xinye.com. Xinye requires signing a [term of access](https://dgraph.xinye.com/clause) before downloading the data, which includes non-commercial restrictions. However, the authors are sharing the dataset publicly via Google Drive. The users may unintentionally violate the term and cause copyright and legal issues. Other datasets may have similar problems.

**Limitations:**

In Section 2, the task definition can be more general. The adjacency matrix may not necessarily be unweighted (e.g., multiple edges between two nodes). Also, the description seems to be transduction. Inductive settings are not considered and are not mentioned in the paper.

In Figure 1, the lower lines of tree ensembles with neighbor aggregation are not significantly higher than other methods. It indicates the instability of the tree ensembles.

**Opportunities For Improvement:**

Although the authors mention classical graph methods in related work, none of them are included in the benchmark. It will be better to evaluate at least one or two.

The authors prove the superiority of tree ensembles with neighbor aggregation, which is parameter-free, but do not include tree ensembles with parametric GNNs. It will be interesting to see whether the GNN+XGBoost can further improve the performance.

**Relation To Prior Work:**

This paper is highly correlated to [1]. While [1] focused on unsupervised settings, this paper benchmarked methods under (semi-)supervised settings. Limited prior work is done under this setting.

[1] Liu, Kay, Yingtong Dou, Yue Zhao, Xueying Ding, Xiyang Hu, Ruitong Zhang, Kaize Ding et al. "Bond: Benchmarking unsupervised outlier node detection on static attributed graphs." *Advances in Neural Information Processing Systems*35 (2022): 27021-27035.

**Summary And Contributions:**

This paper presents a benchmark for supervised graph anomaly detection at the node level on static attributed graphs named GADBench. The benchmark involves 23 models ranging from traditional methods to state-of-the-art GNNs, and 10 diverse real-world datasets. In addition, the authors propose to combine neighborhood aggregation and tree ensemble in the GAD task, which has proven effective in the experiments. Some other insightful findings are discussed in the paper.

---

> ### Author Response · Authors · 2023-08-25
> **Response to Reviewer VEyk - Part 1**
>
> > Q1: Although the authors mention classical graph methods in related work, none of them are included in the benchmark.
>
> R1: We extensively reviewed various classical GAD methods with the intention of integrating them for comparison. However, these methods mentioned in related work either do not utilize labels, overlook node attributes, or have scalability issues. For instance, the BOND benchmark incorporates three classical GAD techniques: SCAN[1], Radar[2], and ANOMALOUS[3]. All of these methods are unsupervised and struggle to process large-scale graphs beyond 20,000+ nodes. Even after modifying ANOMALOUS to enhance its scalability, its performance remained underwhelming. If there is any classical GAD methods that might be a good fit, please let us know, and we'll gladly explore their inclusion.
>
> [1] SCAN: a structural clustering algorithm for networks, in KDD 2007.
> [2] Radar: Residual analysis for anomaly detection in attributed networks, in IJCAI 2017.
> [3] ANOMALOUS: A Joint Modeling Approach for Anomaly Detection on Attributed Networks, in IJCAI 2018.
>
> ---
> > Q2: The authors prove the superiority of tree ensembles with neighbor aggregation, which is parameter-free, but do not include tree ensembles with parametric GNNs. It will be interesting to see whether the GNN+XGBoost can further improve the performance.
>
> R2: Thank you for your suggestion. In the updated version of our paper, we have introduced the BGNN [4] model, which integrates GBDT and GNN into an end-to-end training framework. This integration is realized by allowing new trees to adjust to the gradient updates from GNN. Our empirical analysis indicates that while BGNN excels in specific datasets, it doesn't consistently perform well across all. This inconsistency may result from the inherent instability of its joint training process, especially when compared to the more stable two-step approaches such as XGB-Graph and RF-Graph. We have added this observation in our updated paper and plan to further explore this technique in subsequent research.
>
> [4] Boost then Convolve: Gradient Boosting Meets Graph Neural Networks, in ICLR 2021.
>
> ---
> > Q3: The task definition can be more general. The adjacency matrix may not necessarily be unweighted. Inductive settings are not mentioned in the paper.
>
> R3:  We appreciate your constructive feedback. In our revised paper, we use weighted matrices to represent adjacency matrices and add descriptions related to heterogeneous graphs.  We have also conducted preliminary experiments on heterogeneous graphs and the inductive setting in section 4.2 of the revised paper.
>
> For the inductive setting, we use Dgraph-Fin and Elliptic datasets which have temporal features. In this scenario, features and structures associated with test nodes are not accessible during the training phase. As presented in the following table, the model performance is generally impacted in the inductive setting across both datasets. Specifically, for the Elliptic dataset, XGB-Graph outperforms other models across all metrics in both settings. In contrast, GHRN stands out as the most robust model in the inductive setting for DGraph-Fin.
>
> |    |**Elliptic (Inductive)**|**Elliptic (Transductive)**|**DGraph-Fin (Inductive)**|**DGraph-Fin (Transductive)**|
> |:----|:----|:----|:----|:----|
> |**Model**|AUROC/AUPRC/REC@K|AUROC/AUPRC/REC@K|AUROC/AUPRC/REC@K|AUROC/AUPRC/REC@K|
> |**GCN**|75.79/14.97/16.73|92.40/73.87/69.99|73.99/3.35/5.61|75.85/3.99/7.05|
> |**GraphSAGE**|79.51/19.64/20.59|82.85/34.76/45.95|72.66/3.06/5.43|75.63/3.76/6.97|
> |**BWGNN**|82.29/22.49/28.26|96.12/86.58/81.14|73.85/3.24/5.83|**76.26**/**4.01**/7.52|
> |**GHRN**|84.74/25.42/28.54|96.05/86.57/81.11|**76.20**/**4.03**/**7.48**|76.14/3.99/**7.54**|
> |**XGB-Graph**|**90.36**/**76.20**/**70.64**|**96.80**/**89.58**/**84.59**|71.23/2.81/5.33|74.64/3.66/6.75|

---

> ### Author Response · Authors · 2023-08-25
> **Response to Reviewer VEyk - Part 2**
>
> > Q4: In Figure 1, the lower lines of tree ensembles with neighbor aggregation are not significantly higher than other methods. It indicates the instability of the tree ensembles.
>
> R4: Figure 1 summarizes the results across various datasets, with the lower line specifically representing the performance on DGraph-Fin. As detailed in Tables 4 and 7-9 (10-12 in the revised paper), all methods perform suboptimally on this particular dataset, with XGB/RF-Graph's performance falling in the middle range among all GNNs. This is further explained in section 4.1, where we identify that DGraph-Fin's distinct characteristics, such as high imbalance, sparse structure, and indistinguishable node features, contribute to these results. It's important to note that the perceived instability in Figure 1 is more likely a reflection of the unique challenges presented by the DGraph-Fin dataset, rather than an inherent flaw in the tree ensemble methods themselves.
>
> Furthermore, our analysis of the variance across different runs, presented in Tables 7 and 8, reveals that **XGB-Graph and RF-Graph typically exhibit smaller standard deviations compared to GNN methods**. This evidence indeed supports that the performance of tree ensembles with neighbor aggregation are more stable than GNNs.
>
> ---
> > Q5: The results shown in Table 9 seem problematic. The value in **Ave.** column is not equal to the average of the AUROC for XGBoost.
>
> R5: We appreciate your attention to the details in Tables 8 and 9 (Tables 11 and 12 in our revised paper). The results are correct. To clarify, the **Ave.** column in Table 9 represents the average score of the **first nine datasets**, excluding T-Social, as indicated in the caption. Some comparison methods, such as XGBOD, are unable to complete the random search within the limited time (24 hours). This leads to unstable and unfair random search results on T-Social so we exclude it in calculating average score.
>
> ---
> > Q6: Also, the results are not even better after the hyperparameter search compared to Table 8. The results of other methods may also have similar problems.
>
> R6: This observation doesn't impact the validity of the results in Tables 8 and 9. During the random search process, we save the model that performs best in terms of the AUPRC score on the validation set, and we then report all metrics on the test set. As evidenced in Table 4, this approach generally boosts AUPRC on the test set for all methods. However, **a better AUPRC doesn't always correspond to improved performance across other metrics**.
>
> For instance, consider the training log of BWGNN on the Questions dataset:
>
> ```
> Epoch 176, Loss 0.4428, Val AUROC 0.8041, AUPRC 0.4934, REC@K 0.4961
> Epoch 181, Loss 0.4393, Val AUROC 0.8033, AUPRC 0.4945, REC@K 0.4883
> Epoch 191, Loss 0.4375, Val AUROC 0.8005, AUPRC 0.4954, REC@K 0.4930
> ```
>
> As seen above, an increase in AUPRC can coincide with a decrease in AUROC on the validation set. While saving different models based on different metrics might reduce this effect, we have chosen to consistently use AUPRC to enhance reproducibility.
>
> ---
> > Q7: The memory plot in Figure 1 shows a counterintuitive that CPU memory consumption is usually lower than GPU memory consumption. A more detailed explanation of the measurement of memory consumption is required. In Figure 1 row 4, the title of the y-axis might be AUROC.
>
> R7:  Thank you for pointing out the error in the title of the y-axis in Figure 1 row 4; we have corrected it in the revision.
>
> We monitor CPU/GPU memory consumption through the following code:
>
> ```
> py_process = psutil.Process(os.getpid())
> print(f"CPU Memory Usage: {py_process.memory_info().rss / (1024 ** 3)} GB")
> print(f"GPU Memory Usage: {torch.cuda.memory_reserved() / (1024 ** 3)} GB")
> ```
>
> For instance, when training GAT on the T-Finance dataset, the reported memory usage was:
>
> ```
> CPU Memory Usage: 3.462459564 GB
> GPU Memory Usage: 8.431640625 GB
> ```
>
> While we understand that this measurement might not capture the exact real-time memory usage, we believe it provides a useful relative comparison of memory consumption across different methods. Given that memory usage can be influenced by specific code implementations, the values presented are intended for reference purposes. If there are more accurate or appropriate methods to suggest for this measurement, we're happy to implementing them.

---

> ### Author Response · Authors · 2023-08-25
> **Response to Reviewer VEyk - Part 3**
>
> > Q8: The authors have not clearly explained the difference between heterophily and camouflage mentioned in the paper, which are similar to some extent.
>
> R8: The concepts of heterophily and camouflage, while bearing some similarities, address distinct challenges in graph anomaly detection.
> - Heterophily refers to the **natural variation** between connected nodes, requiring us to manage neighborhood feature dissimilarities in message passing.
> - Camouflage signifies a **deliberate effort** by anomalies to hide themselves by mimicking the normal pattern in the graph, necessitating consideration of intentionally manipulated edges and node features.
>
> We have clarified these distinctions in our revision.
>
> > Q9: The copyright of the dataset.
>
> R9: Upon careful examination of all datasets in GADBench, we identified potential copyright concerns with the DGraph-Fin and Elliptic datasets. We have **removed the processed files related to these datasets and have instead provided a script**, allowing users to preprocess the datasets themselves. We have taken diligent measures to ensure that other datasets are released without such issues. Specifically, Amazon, Yelp, Tolokers, and Questions have been integrated into the DGL library; Weibo and Reddit are accessible through the PyGOD library; T-Social and T-Finance are maintained as part of our previous work. We sincerely appreciate the reviewer for highlighting this matter.

---

> > ### Comment · Reviewer_VEyk · 2023-08-27
> >
> > I appreciate the authors for their detailed revisions, which have substantially addressed the majority of my concerns. Accordingly, I have adjusted my score.

---

### Official Review · Reviewer_eNf8 · 2023-07-20
**a compressive work but improvements are required for a fair conclusion**

**Rating:** 8
**Confidence:** 5
**Correctness:** not really, please see "Opportunities…
**Clarity:** yes

**Strengths:**

Motivation is good.

**Additional Feedback:**

no

**Documentation:**

yes

**Ethics:**

not related

**Limitations:**

yes

**Opportunities For Improvement:**

1. Lack of fair baseline models.

1.1Models based on four categories (Classic Methods, Standard GNNs, Specialized GNNs, and Tree Ensembles with Neighbor Aggregate) were compared, however, there were no methods published in recent three years for Classic Methods, and only one method published in recent three years for Standard GNNs. This is not a fair comparison between these four categories, especially for a survey paper.

1.2  It is known that Ensembles were generally more robust than a single detector. It is not clear why only tree-based ensembles were selected in this paper. Moreover, tree Ensembles with Neighbor Aggregation was actually used in this experiment, but it was not clear the performance of tree Ensembles without Neighbor Aggregation and single detector with Neighbor Aggregation (see the paper https://arxiv.org/abs/2303.09972).

Classic Methods
[12](2001), [15](2011), [17](2016), [18](1967), [66](1958), [85](2018)

Standard GNNs
[30,39,72] (2017), [79,80] (2019),[70] (2021)

Specialized GNNs
[44] (2019),[90] (2020),[32,38,50,77] (2021),[13] (2022),[26] (2023)

Tree Ensembles with Neighbor Aggregate
RF-Graph, XGB-Graph

2. Lack of sufficient explanation of the main conclusion of why the Tree Ensembles with Neighbor Aggregate.

2.1 This explanation was based on visualizing only two datasets after performing t-SNE dimension reduction in section 4.2. However, this was a weak claim based on merely two datasets.

2.2 The roles of ensembles and neighbor aggregate should be analyzed more. Which of them was actually the key reason for their superiority? Because of the diversity of ensembles or the score-smoothing-effect (so that similar nodes should have similar outlier scores as in https://arxiv.org/abs/2303.09972) of the neighbor aggregate, or both? Are they equally important?

3. Lack of necessary reference

3.1 reference of more papers about ensembles

3.2 reference of more papers about neighbor aggregate (see this survey https://ieeexplore.ieee.org/document/10155256)

**Relation To Prior Work:**

yes

**Summary And Contributions:**

Some conclusions were reported in this survey paper based on experiments over serval datasets with serval semi-/supervised models for graph anomaly detection.

---

> ### Author Response · Authors · 2023-08-25
> **Response to Reviewer eNF8 - Part 1**
>
> > Q1: Lack of fair baseline models: there were no methods published in recent three years for classic methods, and only one method published in recent three years for standard GNNs.
>
>
> R1: Our selection criterion for classic methods and standard GNNs is to prioritize the most influential works. While lots of methods have emerged lately, we believe that our chosen baselines have withstood the test of time and continue to be popular, especially in industrial applications.
>
> However, we acknowledge the value of incorporating more recent methods to enhance GADBench. Guided by your suggestion, we've now integrated three recent methods: Neighborhood Averaging (**NA**) [1], Principal Neighbourhood Aggregation (**PNA**) [2], and Boosted Graph Neural Networks (**BGNN**) [3].
>
> We find that the NA model you suggested is a versatile technique adaptable to any model in GADBench. We've integrated this technique into BWGNN as a case study for in-depth analysis. We observed **a remarkable improvement in BWGNN's performance in the semi-supervised setting after incorporating NA**. Specifically, the average AUPRC on 10 datasets improved from 35.4% to 37.0%. However, the boost was not significant in the fully-supervised setting. Based on these findings, we infer that NA offers a potent solution to tackle label scarcity. We've incorporated these insights into our revised paper, and we genuinely appreciate your constructive feedback!
>
> ---
> >Q2: It is known that ensembles were generally more robust than a single detector. It is not clear why only tree-based ensembles were selected in this paper.
>
> R2: Our choice to focus on tree ensembles is due to their reputation as some of the most effective and widely-adopted anomaly detection methods in real applications. For instance, the anomaly detection benchmark ADBench [4] also solely employs tree ensembles among ensemble-based methods and demonstrates their high efficiency and superior performance.
>
> To enrich the diversity of GADBench, we have integrated **BGNN** [3] as a representative of **GNN-based ensemble models**. Preliminary results indicate its ability to surpass most standard GNNs; however, it falls short when compared to XGB-Graph and RF-Graph. A comprehensive discussion on this method is detailed in our revised paper.
>
> ---
> > Q3: (1.2) It was not clear what is the performance of tree ensembles without neighbor aggregation and single detector with Neighbor Aggregation.
> (2.2) The roles of ensembles and neighbor aggregate should be analyzed more: Which of them was actually the key reason for their superiority?
>
> R3:  It's essential to differentiate between the term **neighbor aggregation** in the context of Neighborhood Averaging (**NA**) [1] and in our research due to the different data types involved. In NA, "neighbor" is the group of nearest entities in the feature space (i.e. **feature-based** neighborhood). In contrast, in GADBench, "neighbor" is the directly connected nodes within a given graph (i.e. **structure-based** neighborhood).
>
> In this light, all GNNs we have assessed can be considered as "single detectors with neighbor aggregation". Meanwhile, for tree ensembles without neighbor aggregation, we have evaluated traditional XGBoost and Random Forest models. Our observations, as depicted in Figure 1, 2, and Table 1, highlight that **both tree ensembles and neighbor aggregation are important and their combination can achieve best results.** For instance, Figure 2 shows a consistent enhancement in performance across most datasets when the number of neighbor aggregation layers is increased from 0 (indicating no aggregation) to 2.
>
> ---
> > Q4: Lack of necessary reference about ensembles and neighbor aggregate.
>
> R4: While our initial related work primarily centered around graph anomaly detection methodologies, we have now expanded our references in the revised paper and incorporated discussions on ensemble and neighbor aggregation methods as follows:
>
>  *Beyond trees, various base models can be integrated into ensembles for anomaly detection, as suggested by [6]. The use of neighborhood aggregation has been demonstrated to enhance anomaly detection performance, which can be utilized during various stages such as pre-processing [5], model-training [2], and post-processing phases [1].*
>
> References:
>
> [1] [Neighborhood Averaging for Improving Outlier Detectors, 2023](https://arxiv.org/abs/2303.09972)
>
> [2] [Principal Neighbourhood Aggregation for Graph Nets, 2020](https://arxiv.org/abs/2004.05718)
>
> [3] [Gradient Boosting Meets Graph Neural Networks, 2021](https://openreview.net/pdf?id=ebS5NUfoMKL)
>
> [4] [ADBench: Anomaly Detection Benchmark, 2022](https://arxiv.org/pdf/2206.09426)
>
> [5] [Neighborhood Representative for Improving Outlier Detectors, 2021](https://dl.acm.org/doi/abs/10.1016/j.ins.2022.12.041)
>
> [6] [FOOR: Be Careful for Outlier-Score Outliers When Using Unsupervised Outlier Ensembles, 2023](https://ieeexplore.ieee.org/document/10155256)

---

> ### Author Response · Authors · 2023-08-25
> **Response to Reviewer eNF8 - Part 2**
>
> > Q5: Lack of sufficient explanation of the main conclusion of why the Tree Ensembles with Neighbor Aggregate. This explanation was based on visualizing only two datasets after performing t-SNE dimension reduction in section 4.2. However, this was a weak claim based on merely two datasets.
>
>
> R5:  Thanks for your nice suggestions. We have added the T-Finance and Tolokers datasets to our visualization in Figure 4 of Appendix E (page 20). The observations align with those in Figure 3. Note that the t-SNE algorithm approximates the original data and is not efficient for handling large-scale datasets.
>
> Beyond the insights from the model's decision boundary, we have incorporated an analysis in the updated section 4.3 that delves into the impact of node feature types on model performance. This offers a distinct angle to understand why and when to use tree ensembles with neighbor aggregation.

---

> > ### Comment · Reviewer_eNf8 · 2023-08-25
> > **Thanks for the informative response.**
> >
> > All of my worries were addressed really well, which exceeded my expectations; so, I must raise the score.
> >
> > Given that this work is an excellent study on the influence of neighbor aggregation on graph mining, which is becoming increasingly popular, and the conclusions obtained are important to both academia and industry, I recommend that it be accepted.

---

> > > ### Author Response · Authors · 2023-08-25
> > > **Thank you for your positive feedback.**
> > >
> > > Thank you for your positive feedback and for endorsing our paper. We appreciate the time and effort you dedicated to reviewing our work and engaging in the discussion process. Your valuable insights and suggestions have greatly contributed to the improvement of our paper.

---

> > > > ### Comment · Reviewer_eNf8 · 2023-08-25
> > > > **You are welcome.  Additional action is expected to be taken.**
> > > >
> > > > Since I have generously increased the score, I hope the authors will take action on the two items listed below.
> > > >
> > > > 1. The anomaly score of a node obtained by RF-Graph or XGB-Graph can be further post-processed by NA (simply averaging the node's score with its neighbor nodes' scores to obtain the updated score by NA). It would be great to report the experiment results of "RF-Graph/XGB-Graph +NA". Can NA improve RF-Graph/XGB-Graph? if yes, is the improvement significant or marginal?
> > > >
> > > > 2. Please check your revision carefully and ensure that any of your changes written here are consistent with the blue text in the revision uploaded.

---

> > > > > ### Author Response · Authors · 2023-08-26
> > > > > **Additional results on NA**
> > > > >
> > > > > > It would be great to report the experiment results of "RF-Graph/XGB-Graph + NA". Can NA improve RF-Graph/XGB-Graph? if yes, is the improvement significant or marginal?
> > > > >
> > > > > We examined the impact of introducing NA on the performance of four methods: GIN, BWGNN, XGB-Graph, and RF-Graph. Results are presented below:
> > > > >
> > > > > |           |     Semi-Supervised       |      Fully-Supervised      |
> > > > > |-----------|:-------------------------:|:--------------------------:|
> > > > > |           |     AUROC/AUPRC/Rec@K      |     AUROC/AUPRC/Rec@K       |
> > > > > | GIN       |     73.61/30.26/32.14     |     79.97/36.54/37.52      |
> > > > > | + NA   |     **74.78/33.91/34.18**     |     **80.43/39.03/40.01**      |
> > > > > | BWGNN     |     76.00/35.42/**37.36**     |     **84.73**/**48.19**/48.01      |
> > > > > | + NA   |     **76.95/36.96**/37.34     |     84.70/47.93/**48.08**      |
> > > > > | XGB-Graph |     77.74/42.91/41.29     |     **86.71**/**61.09**/**57.82**      |
> > > > > | + NA   |     **78.41**/**44.30**/**42.07**     |     86.05/59.58/57.41      |
> > > > > | RF-Graph  |     78.94/43.37/40.83     |     **85.35**/**59.65**/**56.90**      |
> > > > > | + NA   |     **79.56**/**45.14**/**42.28**     |     85.17/58.54/55.33      |
> > > > >
> > > > > Our key observations are:
> > > > >
> > > > > - In the **semi-supervised** setting, all four methods show enhanced performance after integrating NA. For example, **the AUPRC score is improved by 1.4%~3.7%**. Given these results are averaged over **100 trials** (10 runs on10 datasets), we believe  **this improvement is significant**.
> > > > >
> > > > > - In the **fully-supervised** setting, NA can enhance weaker baselines like GIN. However, for stronger baselines, the performance **remains similar** (on BWGNN) or might even **decrease** (on XGB-Graph and RF-Graph).
> > > > >
> > > > > Our evaluation still has certain limitations due to time constraints. We utilized a fixed neighborhood number of 5 across all datasets. A more flexible neighborhood number might be better. We will continue to investigate NA as well as other pre-processing and post-processing techniques. We aim to make these techniques **accessible for all models in GADBench**, providing users a broader spectrum of options.
> > > > >
> > > > > ---
> > > > > > Please check your revision carefully and ensure that any of your changes written here are consistent with the blue text in the revision uploaded.
> > > > >
> > > > > Thank you for pointing this out. In our earlier version of the revision, we inadvertently omitted a paragraph on related work. We have corrected this and will continue to thoroughly review and refine our manuscript.

---

> > > > > > ### Comment · Reviewer_eNf8 · 2023-08-26
> > > > > > **thanks for the information**
> > > > > >
> > > > > > I appreciate the authors' response.
> > > > > >
> > > > > > The findings are interesting as they uncovered how the neighbor aggregation in structure/feature space and score space are complementary to each other in semi-supervised vs. supervised settings. Readers should expect to see these results in the final version. Hence, the authors are strongly encouraged to add these results and observations to the final version.
> > > > > >
> > > > > > Good luck with the acceptance!

---

> > > > > > > ### Author Response · Authors · 2023-08-29
> > > > > > > **Author response to Reviewer eNF8**
> > > > > > >
> > > > > > > Certainly. We will evaluate NA on most models within GADBench and incorporate the findings into the final version.

---

### Official Review · Reviewer_qCKR · 2023-07-21

**Rating:** 7
**Confidence:** 4
**Correctness:** Yes
**Clarity:** Yes

**Strengths:**

The paper is highly original. It presents a comprehensive benchmark for supervised anomalous node detection on static graphs, which is an important and challenging problem in the field of graph analysis. The authors provide a thorough evaluation of traditional Graph Anomaly Detection algorithms and Graph Neural Networks, and compare their performance and efficiency on large-scale graphs. The significance of the paper is high, as it provides important insights into the current advancements of GAD and establishes a solid foundation for future research.

**Additional Feedback:**

- Have you considered using other types of graph anomalies, such as edge anomalies or subgraph anomalies, in addition to node anomalies?
- For the dataset description table (the first table on page 5), you can add the information about which datasets contain which types of features, and whether temporal features are present in each of the datasets.

**Documentation:**

Yes

**Limitations:**

Yes

**Opportunities For Improvement:**

The paper focuses only on supervised anomalous node detection on static graphs, and does not consider other types of graph anomalies or dynamic graphs.

**Relation To Prior Work:**

Yes

**Summary And Contributions:**

The paper presents GADBench, a comprehensive benchmark for supervised anomalous node detection on static graphs. The authors revisit and benchmark traditional Graph Anomaly Detection algorithms and GNNs to determine their performance and efficiency on large-scale graphs. The authors make GADBench available as an open-source tool.

---

> ### Author Response · Authors · 2023-08-25
> **Response to Reviewer qCKR**
>
> > Q1: The paper focuses only on supervised anomalous node detection on static graphs, and does not consider other types of graph anomalies or dynamic graphs. Have you considered using other types of graph anomalies, such as edge anomalies or subgraph anomalies?
>
> R1: In the current scope of GADBench, we primarily focused on node anomalies. The consideration of other types of graph anomalies, such as edge anomalies or subgraph anomalies, is indeed valuable and can provide a more comprehensive view of the graph's structure and behavior. We appreciate your insight and plan to extend our research to encompass these anomaly types in future iterations of GADBench.
>
> Regarding dynamic graphs, in GADBench, only DGraph-Fin and Elliptic datasets have temporal information, which is embodied as node features and influences the train-test split. Deploying GAD methods for dynamic graphs might result in biased comparisons for other 8 datasets without temporal attributes. Furthermore, introducing new dynamic graph datasets to GADBench might also compromise the evaluation of the established baselines. As a result, we focus on static graphs in this version.
>
> To enhance GADBench, we have added experiments in an **inductive setting** with the DGraph-Fin and Elliptic datasets, both of which possess temporal features. In this scenario, features and structures associated with test nodes are not accessible during the training phase, akin to a simplified version of time-dependent anomaly detection. Additionally, we have shared preliminary results on heterogeneous graphs. Detailed results and analysis are provided in **Section 4.2** of the revised paper.
>
> ---
> > Q2: For the dataset description table (the first table on page 5), you can add the information about which datasets contain which types of features, and whether temporal features are present in each of the datasets.
>
> R2: Thank you for the valuable suggestion. We've incorporated a new column in **Table 3** and provided an extended table in **Appendix B** of the revised paper. These additions present the specific node feature types in each dataset, clearly indicating the presence of temporal features where applicable. We also discussed the impact of dateset feature types on model performance in the updated **Section 4.3**.
>
> |Dataset|Feature Type|Detailed Feature Description|Feature Dimension|
> |:----|:----|:----|:----|
> |Reddit|Text Embedding|LIWC text embedding for posts|64|
> |Weibo|Text Embedding|Bag-of-words features from posts|400|
> |Amazon|Miscellaneous|Hand-crafted user features and statistics|25|
> |YelpChi|Miscellaneous|Hand-crafted review features and statistics|32|
> |Tolokers|Miscellaneous|User profile with task performance statistics|10|
> |Questions|Text Embedding|FastText embeddings for user descriptions|301|
> |T-Finance|Miscellaneous|User profile details, including registration days and logging activities|10|
> |Elliptic|Miscellaneous|**Timestamps** and transaction information|166|
> |DGraph-Fin|Miscellaneous|**Timestamps** and user profiles with details such as age and gender|17|
> |T-Social|Miscellaneous|User profile details, including registration days and logging activities|10|

---

> > ### Comment · Reviewer_qCKR · 2023-08-25
> >
> > The author has addressed most of my concerns. Thus I have raised my score to 7

---

> > > ### Author Response · Authors · 2023-08-25
> > > **Thank you for your positive feedback.**
> > >
> > > Thank you for your positive feedback and for endorsing our paper. We appreciate the time and effort you dedicated to reviewing our work and engaging in the discussion process. Your valuable insights and suggestions have greatly contributed to the improvement of our paper.

---

### Official Review · Reviewer_jFM3 · 2023-07-25
**Some concerns should be addressed**

**Rating:** 6
**Confidence:** 4
**Correctness:** Please see the Limitations.
**Clarity:** Please see the Limitations.

**Strengths:**

1.  The finding that a tree ensemble with neighborhood aggregation can deliver the best performance is interesting and worth further investigations.

2. GADBench is available as an open-source tool, which can help researchers test their model and keep track of current research frontier.

**Additional Feedback:**

Please see the Limitations.

**Documentation:**

Yes.

**Ethics:**

None.

**Limitations:**

1. Lack of baseline models. As this paper is a benchmark for GAD models, I would expect a comprehensive evaluation of existing models. But I find that this benchmark lacks some popular baseline models in the experiment, like Care-GNN[1], GraphConsis[2], FRAUDRE [3], H2-FDetector [4], to name a few.

2. Findings lack of novelty. I find that some of the findings in section 4.1 is not informative. For example, the finding that most standard GNNs prove unsuitable for GAD is repeatedly mentioned in existing GAD works. Also, the finding that Specialized GNNs require hyperparameter tuning to achieve satisfactory performance is a common sense in the machine learning community. I would expect authors can derive more valuable findings based on experiments in this benchmark.

3. Simplification of dataset causes information loss. Authors simply treat all datasets as static homogeneous graphs, ignoring temporal and heterogeneous information. That can bias the testing results as some baselines are specifically designed to utilize these kinds of information. In this way, we cannot get a fair evaluation on these models.

[1] Yingtong Dou, et al. 2020. Enhancing graph neural network-based fraud detectors against camouflaged fraudsters. CIKM.

[2] Liu, Zhiwei, et al. 2020. Alleviating the inconsistency problem of applying graph neural
network to fraud detection. SIGIR.

[3] Ge Zhang, et al. 2021. FRAUDRE: fraud detection dual-resistant to graph inconsistency
and imbalance. ICDM.

[4] Fengzhao Shi, et al. 2022. H2-FDetector: a gnn-based fraud detector with homophilic and heterophilic connections. WWW.

**Opportunities For Improvement:**

Please see the Limitations.

**Relation To Prior Work:**

Please see the Limitations.

**Summary And Contributions:**

This work proposes GADBench, which is a benchmark for supervised anomalous node detection on static attributed graphs. It evaluates 23 GAD methods across 10 real-world datasets in semi-supervised and fully-supervised settings. The authors find that a tree ensemble with simple neighborhood aggregation can outperform all other baselines and provide an initial study on reasons behind that.

---

> ### Author Response · Authors · 2023-08-25
> **Response to Reviewer jFM3 - Part 1**
>
> > Q1: This benchmark lacks some popular baseline models in the experiment, like Care-GNN[1], GraphConsis[2], FRAUDRE [3], H2-FDetector [4], to name a few.
>
> R1: Thank you for pointing out the absence of popular baseline models like Care-GNN, GraphConsis, FRAUDRE, and H2-FDetector in our benchmark. We recognize the significance of these methods in the domain. However, it's worth noting that the primary design of these methods is geared towards heterogeneous graphs.  All these works only conducted experiments on the Amazon and YelpChi datasets in their original paper. The direct application of these models to homogeneous graphs, which consists 8 out of 10 datasets in GADBench, would necessitate considerable alterations in both model architecture and codebase. Such adaptations could compromise the original performance that these models offer.
>
> Nonetheless, given the potential interest from the community in understanding these heterogeneous GAD methods, we've taken a step forward based on your feedback. Our revised paper now includes a new section 4.2 comparing the performance of these heterogeneous methods with other methods in GADBench on Amazon and YelpChi. Given the limited timeframe of the rebuttal phase, we've integrated evaluations for Care-GNN and GraphConsis. We are optimistic about incorporating FRAUDRE and H2-FDetector into the GADBench repository in the near future.
>
> As illustrated in the following table, accounting for heterogeneity doesn't improve performance on the Amazon dataset. However, it does yield benefits on the Yelp dataset. Specifically, in the semi-supervised setting, CARE-GNN significantly surpasses GAT, BWGNN, and XGB-Graph. Yet, in the fully-supervised setting, XGB-Graph remains the best.
>
> |    |Amazon(Semi-Supervised)|Amazon(Fully-Supervised)|Yelp(Semi-Supervised)|Yelp(Fully-Supervised)|
> |:----|:----|:----|:----|:----|
> |Model|AUROC/AUPRC/REC@K|AUROC/AUPRC/REC@K|AUROC/AUPRC/REC@K|AUROC/AUPRC/REC@K|
> |GraphConsis|91.96/76.14/73.28|95.17/82.36/77.39|87.08/49.69/50.32|93.31/69.28/64.43|
> |CAREGNN|86.00/58.95/59.12|90.84/72.64/67.72|**91.19/68.69/65.35**|95.23/81.06/74.85|
> |GAT|92.44/81.57/77.07|96.66/86.67/83.10|65.56/25.03/28.08|79.50/43.41/43.65|
> |BWGNN|91.83/81.68/77.71|97.95/89.09/85.00|64.30/23.66/26.44|84.89/55.06/52.18|
> |XGB-Graph|**94.68/84.38/78.17**|**98.69/92.61/85.87**|64.03/24.84/26.81|**96.22/87.03/78.76**|
>
> ---
>
> > Q2: I would expect authors can derive more valuable findings based on experiments in this benchmark.
>
> R2: We appreciate the reviewer's feedback on the novelty of some findings in section 4.1. Although many studies have explored comparative analyses of various GNNs, our research is uniquely positioned to compare GNNs with different model categories, notably tree-based models. Consequently, our primary finding, the effectiveness of tree ensembles with neighbor aggregation, introduces a novel perspective that stands apart from previous work.
>
> To further enrich our study, we have added the following novel findings in the revised paper:
> - In section 4.2, we have embarked on preliminary explorations into heterogeneous graphs and the inductive setting, further enriching our benchmark's scope.
> - In section 4.3, we uncovered relationships between the node feature types and model performance, providing initial guidance for users to determine whether they should prioritize GNNs or ensemble trees on a particular dataset.
>
> We are confident that these additions will provide more valuable insights to readers and further strengthen the contribution of our work.

---

> ### Author Response · Authors · 2023-08-25
> **Response to Reviewer jFM3 - Part 2**
>
> > Q3: Authors simply treat all datasets as static homogeneous graphs, ignoring temporal and heterogeneous information. In this way, we cannot get a fair evaluation on these models.
>
> R3: The topic of heterogeneity was addressed in R1. As for treating datasets as static graphs, we recognize that only DGraph-Fin and Elliptic in GADBench incorporate temporal information. This temporal aspect is embodied as node features and influences the train-test split. Similar to the challenge with heterogeneous graphs, deploying dynamic GAD methods might result in biased comparisons for other 8 datasets without temporal attributes.
>
> To offer a holistic evaluation, we conduct experiments in the inductive setting using Dgraph-Fin and Elliptic datasets which have temporal features. In this scenario, features and structures associated with test nodes are not accessible during the training phase, akin to a simplified version of time-dependent anomaly detection. As presented in the following table, the model performance is generally impacted in the inductive setting across both datasets. Specifically, for the Elliptic dataset, XGB-Graph outperforms other models across all metrics in both settings. In contrast, GHRN stands out as the most robust model in the inductive setting for DGraph-Fin.
>
> |    |**Elliptic (Inductive)**|**Elliptic (Transductive)**|**DGraph-Fin (Inductive)**|**DGraph-Fin (Transductive)**|
> |:----|:----|:----|:----|:----|
> |**Model**|AUROC/AUPRC/REC@K|AUROC/AUPRC/REC@K|AUROC/AUPRC/REC@K|AUROC/AUPRC/REC@K|
> |**GCN**|75.79/14.97/16.73|92.40/73.87/69.99|73.99/3.35/5.61|75.85/3.99/7.05|
> |**GraphSAGE**|79.51/19.64/20.59|82.85/34.76/45.95|72.66/3.06/5.43|75.63/3.76/6.97|
> |**BWGNN**|82.29/22.49/28.26|96.12/86.58/81.14|73.85/3.24/5.83|**76.26**/**4.01**/7.52|
> |**GHRN**|84.74/25.42/28.54|96.05/86.57/81.11|**76.20**/**4.03**/**7.48**|76.14/3.99/**7.54**|
> |**XGB-Graph**|**90.36**/**76.20**/**70.64**|**96.80**/**89.58**/**84.59**|71.23/2.81/5.33|74.64/3.66/6.75|

---

> > ### Author Response · Authors · 2023-08-29
> > **Follow-up response to Reviewer jFM3**
> >
> > We've successfully implemented **H2-FDetector** and integrated it into the GADBench GitHub repository. The results are very close to the original paper. Due to limited time, a comparative analysis of this method will be included in our final version.

---

### Official Review · Reviewer_7otD · 2023-07-28
**Review of GADBench: Revisiting and Benchmarking Supervised Graph Anomaly Detection**

**Rating:** 6
**Confidence:** 4
**Correctness:** Yes.
**Clarity:** Yes.

**Strengths:**

+ This paper formalizes a very interesting problem, i.e., graph anomaly detection.

+ The paper is well-written and easy to follow.

+ The datasets covered in this paper include 10 diverse topics.

+ The authors provide complete experiments and analysis.


**Additional Feedback:**

No.

**Documentation:**

No ethical and responsible use.

**Ethics:**

No ethics mentioned in the paper.

**Limitations:**

The experimental results are also well described and convincing. However, the authors are suggested to consider more types of datasets and give more motivations regarding datasets selection.

**Opportunities For Improvement:**

- It would be great if the authors can also add experimental results over dynamic graphs, i.e., time-dependent graph anomaly detection.

- The authors may need to explain why they consider these 10 diverse datasets. Considering it is a "Datasets and Benchmarks" track, dataset is not novel enough and I suggest the authors provide more motivations about datasets selection (for the experiment, I think it is very comprehensive now).

**Relation To Prior Work:**

Yes.

**Summary And Contributions:**

This paper presents a GADBench, i.e., a comprehensive benchmark for supervised anomalous node detection on static graphs. Specifically, the proposed GADBench provides a thorough comparison across distinct models on ten real-world GAD datasets ranging from thousands to millions of nodes.

---

> ### Author Response · Authors · 2023-08-25
> **Response to Reviewer 7otD**
>
> > Q1: It would be great if the authors can also add experimental results over dynamic graphs, i.e., time-dependent graph anomaly detection.
>
> R1: In GADBench, only DGraph-Fin and Elliptic datasets have temporal information, which is embodied as node features and influences the train-test split. Implementing methods designed for dynamic graphs could lead to skewed comparisons when dealing with datasets that lack temporal features. Introducing new dynamic graph datasets to GADBench might also compromise the evaluation of the established baselines. As a result, we focus on static graphs.
>
> To enhance the utility of GADBench, we have added experiments in an inductive setting with the DGraph-Fin and Elliptic datasets, given their temporal attributes. In this scenario, the features and structures associated with test nodes are not accessible during the training phase, akin to a simplified version of time-dependent anomaly detection. Besides, we have shared preliminary results on another type of dataset---heterogeneous graphs. Detailed results and analysis are provided in section 4.2 of the revised paper.
>
> > Q2: The authors may need to explain why they consider these 10 diverse datasets. Considering it is a "Datasets and Benchmarks" track, dataset is not novel enough and I suggest the authors provide more motivations about datasets selection.
>
> R2: In section 3.2 of our paper, we detailed the criteria guiding our dataset selection for GADBench. Specifically, we collected datasets from **diverse domains, ensuring the presence of organic anomalies, varied relationships, scalability considerations, and realistic anomaly proportions**. Our main goal is to **align GAD research with actual real-world challenges**. While we believe our selection is comprehensive, we recognize room for expanding dataset diversity and will consider more types in subsequent updates.

---

### Author Response · Authors · 2023-08-25
**Summary of Our Responses**

We deeply appreciate the reviewers for their constructive and insightful feedback. We have carefully reviewed each comment and addressed them with detailed, point-by-point responses. We upload the revised paper with changes highlighted in blue. Additionally, we have included a single PDF that combines the main content with the appendix, in the Supplementary Material.

The primary changes are summarized below:

- **New baselines.**  Five additional baselines have been incorporated into GADBench. These include NA and PNA as recommended by reviewer eNf8, CARE-GNN and GraphConsis from reviewer jFM3, and BGNN from reviewer VEyk. GADBench now encompasses 28 methods, and we plan to continually expand this.

- **Additional experiments.** We've conducted new experiments on heterogeneous graphs as suggested by reviewer jFM3 and inductive settings as proposed by both reviewers jFM3 and VEyk. For detailed results and analysis, kindly refer to section 4.2 of the revised paper, as well as our responses R1 and R3 to reviewer jFM3.

- **In-depth analaysis.** At the suggestion of reviewer eNf8, we've incorporated two more datasets for decision boundary analysis. Additionally, a fresh perspective on the correlation between dataset feature types and model performance has been included in section 4.3.

- **Other points.** We have refined the task definition based on feedback from reviewer 7otD. We have also incorporated more related works, as suggested by reviewers eNf8 and jFM3. We have updated Table 3 and introduced Table 7 in Appendix B to summarize the feature types of datasets in GADBench, according to reviewer qCKR.

We sincerely thank all reviewers for taking the time to read our response. Please let us know if our revisions sufficiently address your concerns. We are always available for further clarifications during the discussion phase.

---

### Decision · Program_Chairs · 2023-09-22

**Decision:**

Accept (Poster)

**Comment:**

Thank you for submitting your paper to NeurIPS.

The reviewers find your proposed datasets valuable for important applications. They appreciate the extensiveness of both the datasets and the experiments conducted, as well as your efforts to make the datasets easily accessible.

However, the reviewers have raised some concerns, which have not been fully addressed during the rebuttal period:
- Comparison with additional competitors
- Discussion of additional relevant studies.

Overall, all reviewers have a positive view of the paper and lean towards its acceptance. I believe that the additional concerns can be adequately addressed in the camera-ready version. Therefore, I recommend accepting the paper.